# How Much Cryosphere Model Complexity is Just Right? Exploration Using the Conceptual Cryosphere Hydrology Framework

Thomas M. Mosier[1,2], David F. Hill[1,3], and Kendra V. Sharp[1,2]

[1]Water Resources Graduate Program, Oregon State University, Corvallis, Oregon, USA 97331
[2]Mechanical, Industrial, and Manufacturing Engineering, Oregon State University, Corvallis, Oregon, USA 97331
[3]Civil and Construction Engineering, Oregon State University, Corvallis, Oregon, USA 97331

*Correspondence to:* mosier.thomas@gmail.com

**Abstract.** Making meaningful projections of the impacts that possible future climates would have on water resources in mountain regions requires understanding how cryosphere hydrology model performance changes under altered climate conditions and when the model is applied to ungaged catchments. Further, if we are to develop better models, we must understand which specific process representations limit model performance. This article presents a modeling tool, named the Conceptual Cryosphere Hydrology Framework (CCHF), that enables implementing and evaluating a wide range of cryosphere modeling hypotheses. The CCHF represents cryosphere hydrology systems using a set of coupled process modules that allows easily interchanging individual module representations and includes analysis tools to evaluate model outputs. CCHF version 1 implements model formulations that require only precipitation and temperature as climate inputs - for example variations on simple degree-index (SDI) or enhanced temperature index (ETI) formulations - because these model structures are often applied in data-sparse mountain regions, and perform relatively well over short periods, but their calibration is known to change based on climate and geography. Using CCHF, we implement seven existing and novel models, including one existing SDI model, two existing ETI models, and four novel models that utilize a combination of existing and novel module representations. The novel module representations include a heat transfer formulation with net longwave radiation and a snowpack internal energy formulation that uses an approximation of the cold content. We assess the models for the Gulkana and Wolverine glaciated watersheds in Alaska, which have markedly different climates and contain long-term US Geological Survey benchmark glaciers. Overall we find that the best performing models are those that are more physically consistent and representative, but that no single model performs best for all of our model evaluation criteria.

# 1 Introduction

Robustly understanding connections between the climate, cryosphere, and streamflow is necessary to make informed decisions regarding water resources in mountainous regions. Commonly, distributed hydrologic models are applied to assess impacts of climate change (i.e. climatic conditions for which the model was not calibrated) or ungaged basins (i.e. geographies for which the model was not calibrated). While energy balance models are theoretically more robust under altered climate and geographic settings, conceptual models are often implemented for data-sparse mountain regions (e.g. Hagg et al., 2007; Jung and Chang, 2011; Lutz et al., 2014) because they require fewer input data than energy balance models and are often considered "good enough". Further, because energy balance models require less frequently measured inputs, it is typically not known, or studied, how uncertainty in the additional required inputs propagates through the model and impacts results. Therefore, an important question is, how robust are conceptual cryosphere hydrology models under geographies and climates for which the models were not calibrated? And, are there novel conceptual formulations that are more robust than existing conceptual models? Our present work attempts to address these topics.

Our basic definition of the difference between conceptual and energy balance models is that energy balance models represent each heat flux independently and base calculation of heat fluxes on a much larger set of physically relevant parameters, while conceptual models lump multiple heat fluxes together and use parameterizations to reduce the number of required input variables. Specifically, the only climatic variables typically used in conceptual models are air temperature and precipitation. In contrast, energy balance models often require additional variables such as wind speed, relative humidity, and air pressure.

In reality, conceptual and energy balance models exist on a spectrum rather than as entirely distinct model categories because every model is an abstraction of reality and even energy balance models make significant assumptions about the relevant processes. Simultaneously, it's probable that improvements to conceptual model formulations will stem from rooting the conceptual formulation more firmly in the energy balance. The reason that energy balance models are theoretically more robust than conceptual models is that the relative balance of heat fluxes changes over time and between locations (Kustas et al., 1994; Sicart et al., 2008; Huss et al., 2009). Several of the energy balance inputs are even more sparsely measured in mountain environments than precipitation and temperature and often vary significantly over short spatial and temporal scales (see discussion on the surface energy balance in Sturm, 2015). As an example, wind speed linearly scales each convective heat flux (i.e. sensible and latent), and is therefore necessary for energy balance models. The spatial distribution of wind speed is, however, difficult to characterize even if there are measurements of wind speed at points within the region (Marks et al., 1992; Marks and Winstral, 2001). Therefore, it warrants consideration of whether including variables such as wind speed improves or reduces model performance in a given application.

One advantage that conceptual models may have, therefore, over energy balance models in mountain environments is that they require fewer inputs and these inputs may have lower uncertainties (despite the uncertainties still being higher for mountain regions than for adjacent lowlands; Hijmans et al., 2005). Conceptual models perform well under many circumstances (Ohmura, 2001), with enhanced temperature index (ETI) models, which base melt magnitude on shortwave radiation and temperature, typically performing better than simple degree index (SDI) models, which base melt magnitude only on temperature

(Hock, 1999; Pellicciotti et al., 2005). The reasons for relatively good performance of SDI and ETI models are that many heat fluxes are functions of temperature (e.g. sensible convection and longwave radiation), seasonality of air temperature and shortwave radiation are often correlated for a given location, and shortwave radiation fluxes tend to be significantly larger than convective heat fluxes during melt conditions (Marks et al., 1992; Cline, 1997). While previous studies have compared ETI to

energy balance models (e.g. Hock, 2005), the present study seeks to assess whether adding more physical basis to conceptual cryosphere structures (while holding the required inputs constant) enhances model accuracy and precision.

For model performance to be robust, the means by which a model calculates output must be representative of the entire range of physical conditions being modeled (both climatic regimes and geographies). Encapsulated in the above concept is equifinality, which is specifically defined as the existence of multiple distinct parameter sets for a given model that perform

equally well during model calibration (Beven, 2006). The existence of multiple parameter sets that equally explain the observations used in evaluation is problematic because each parameter set corresponds to a different set of assumptions about the physical system and may behave differently under perturbations to the physical system (i.e. applications for different geographies and climates). Including fitting parameters in a model is not inherently problematic though, because fitting parameters represent modeled approximations to the physical system and may allow both better calibration and validation performance,

provided that the calibration criteria are sufficient to minimize equifinality and adequately represent the dominant processes being modeled (Konz et al., 2010; Finger et al., 2015; Her and Chaubey, 2015).

Selecting suitable evaluation criteria is therefore important, particularly when a model is to be calibrated for a relatively small area and then applied over a much larger region (Arsenault and Brissette, 2014). In general, longer model evaluation periods reduce equifinality because longer periods increase the likelihood that a diverse set of physical events will be included

in the evaluation (Razavi and Tolson, 2013; Her and Chaubey, 2015); however, using multi-objective evaluation criteria can also be important for reducing equifinality (Finger et al., 2011, 2015; Silvestro et al., 2015). In the case of cryosphere hydrology models, a common set of evaluation criteria are streamflow, snow covered area, and glacier stakes (Ragettli and Pellicciotti, 2012; Finger et al., 2015). Some multi-objective evaluation schemes calibrate model parameters in a single stage (e.g. Finger et al., 2015) and others utilize multiple stages (e.g. Ragettli and Pellicciotti, 2012, use three stages). In multi-stage calibration,

it is important to consider the hierarchy of processes, i.e. how modeled processes depend on one another, and construct the calibration to reflect these relationships.

Identifying improvements to conceptual models requires a framework for comparing existing structures and systematically assessing novel representations. Other hydrology intercomparison projects have been carried out (e.g. Smith et al., 2004, 2012; Clark et al., 2008) and are being developed (e.g. Clark et al., 2015a, b). The comparison carried out by Smith et al. (2004) and

Smith et al. (2012) assesses several hydrologic models, many of which are conceptual, but implement each model independently instead of through a common framework. Clark et al. (2008) create a common modeling structure for implementing lumped variants of several hydrologic models for two basins in the USA, but do not model snowpack accumulation or melt. The Structure for Unifying Multiple Modeling Alternatives (SUMMA) is a much more generalizable intercomparison framework by Clark et al. (2015a). In SUMMA, the relevant mass and energy conservation relationships are implemented, and the structure

is designed to allow alternative spatial representations of each process relevant to the system. Thus, in theory SUMMA can

be developed to implement any physically-based representation of a distributed hydrologic system. In practice, SUMMA development has not focused on the processes and conditions typical of data-sparse mountain regions (Clark et al., 2015a, b).

The overall goal of our present work is to assess and improve conceptual representations of snow and glacier processes. To achieve this, we have developed the Conceptual Cryosphere Hydrology Framework (CCHF). The CCHF modularizes alpine and glacier surface processes, and includes routines to optimize model parameters and analyze outputs. The specific objectives of this study are to (1) demonstrate that CCHF is a useful tool for developing novel conceptual cryosphere hydrology models and (2) explore differences in accuracy and precision of existing and novel conceptual cryosphere hydrology models for two well-monitored glaciated model domains. We assess four heat flux representations (including three existing and one novel formulation) and two mass flux representations (one existing and one novel). We focus on these processes because their representations may depend significantly on climate regime. We conduct our assessment for Gulkana and Wolverine watersheds in Alaska (Fig. 1), for the period July 2000 through June 2010, and evaluate model performance using streamgage, glacier stake, and snow-covered area (SCA) observations. We assess each model through applying it to the opposite watershed for the same period, which enables us to evaluate the models for a long period using all three types of observation data and serves as a test of how well the models perform under altered geographies and climates. In addition to our present assessment, a valuable attribute of CCHF is that it is open-source (distributed on GitHub under 'thomasmosier/CCHF') and can easily be implemented for other regions, different climate forcing datasets, or novel process representations.

## 1.1 Existing Conceptual Cryosphere Heat and Melt Representations

A basic, yet still widely used, conceptual cryosphere heat and melt formulation is the SDI structure (two well known implementations are the Hydrologiska Byråns Vattenbalansavdelning, presented in Lindström et al., 1997, and the Snowmelt-Runoff Model, presented in Martinec, 1975). In SDI models, melt, $M$ ($[M] =$ m, which represents depth of water equivalent melt per unit area; throughout, square brackets denote a variable's units), occurs when the surface air temperature at a grid cell, $T_a$ ($[T_a] = {}^\circ$C), is above a temperature threshold, $T_0$ ($[T_0] = {}^\circ$C) (Singh et al., 2000; Wang et al., 2004). Otherwise, no melt occurs and incident precipitation during that time step adds to the snowpack. This is represented mathematically as

$$M = \begin{cases} f_m \left(T_a - T_0\right) \Delta t & \text{if } T_a > T_0 \\ 0 & \text{otherwise} \end{cases} \tag{1}$$

where $f_m$ is the degree-index factor ($[f_m] =$ m ${}^\circ$C${}^{-1}$ s${}^{-1}$) and $\Delta t$ is the model time step ($[\Delta t] =$ s). In some SDI models the value of $f_m$ is taken to be the same for snow and ice melt (e.g. Wang et al., 2004) and other model implementations use different values of $f_m$ for snow and ice melt (e.g. Singh et al., 2000).

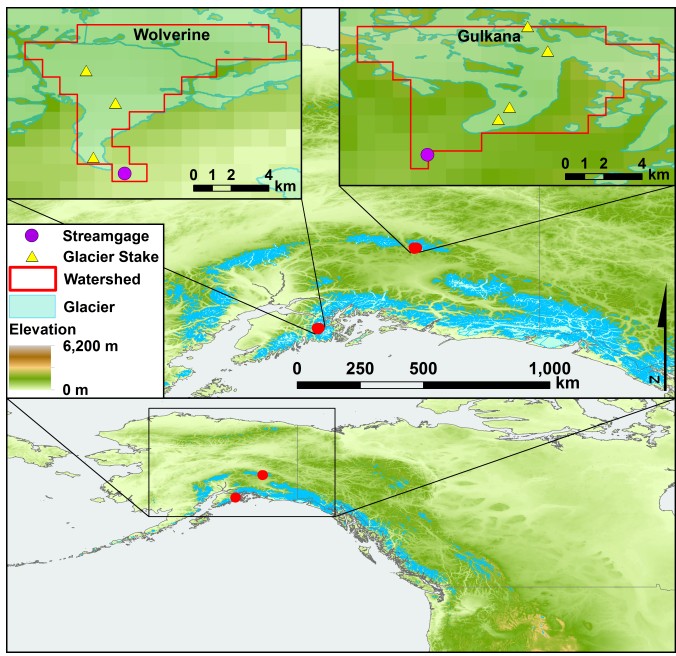

**Figure 1.** Gulkana and Wolverine model domains are depicted using insets in the upper right and upper left, respectively. While the model is implemented for the entire rectilinear region, the red border highlights the grid cells contributing to streamflow at the US Geological Survey (USGS) streamgages in each domain. The digital elevation model is produced by WorldClim (Hijmans, 2011) and glacier coverage is from the Randolph Glacier Inventory version 5 (Pfeffer et al., 2014).

ETI formulations expand on SDI structures by including shortwave radiation, $I$ ($[I] = $ W m$^{-2}$). Two ETI formulations, respectively by Hock (1999) and Pellicciotti et al. (2005), are

$$M = \begin{cases} \left(a_{s/i}\, I + f_m\right)\left(T_a - T_0\right)\Delta t & \text{if } T_a > T_0 \\ 0 & \text{otherwise} \end{cases} \tag{2}$$

$$M = \begin{cases} \left(a_p\left(1 - \alpha\right) I + f_m\left(T_a - T_0\right)\right)\Delta t & \text{if } T_a > T_0 \\ 0 & \text{otherwise} \end{cases} \tag{3}$$

5   where $a_{s/i}$ is a fitting parameter that takes different values for snow melt versus ice melt ($[a_{s/i}] = $ m$^3$ J$^{-1}$ $^\circ$C$^{-1}$), $a_p$ is a fitting parameter that scales melt from shortwave radiation ($[a_p] = $ m$^3$ J$^{-1}$), and $\alpha$ is the surface albedo (unitless). Note that the coefficient symbols and units have been changed from those used in the citations in order to simplify the model representations and ensure consistent notation throughout this article.

    These existing conceptual melt formulations (Eqs. 1-3) use a step function to calculate when melt conditions occur. Both

10  SDI and ETI models typically ignore internal energy of the snow or ice. The primary implication of this simplification is that these conceptual models may not accurately capture the timing of melt or freezing onset. Many conceptual cryosphere model

implementations avoid this potential issue by focusing only on the melt season (e.g. Kustas et al., 1994, Hock, 1999, Pellicciotti et al., 2005, and Sicart et al., 2008).

## 1.2 Cryosphere Energy Balance Representations

The existing conceptual cryosphere models described in Eqs. 1-3 combine the the snow and ice heat flux, internal energy, and mass flux into a single simplified formulation. The premise of these conceptual representations is that temperature is a major determinant of the energy balance (temperature affects convective heat transfer, longwave radiation, and conduction) and that surface melt typically occurs when the air temperature is at some threshold above freezing. In reality, the net heat flux modifies the sensible and latent energy of the snowpack or ice, which can diverge from the temperature threshold assumption in Eqs. 1-3. While there are different methods for formulating the surface energy balance, the representation used in Hock (2005) is

$$Q_N + Q_H + Q_L + Q_R + Q_G + Q_M = 0 \tag{4}$$

where $Q_N$ is the net radiation, $Q_H$ is the sensible heat flux, $Q_L$ is the latent heat flux ($Q_H$ and $Q_L$ are sometimes referred to as the turbulent heat fluxes), $Q_R$ is sensible heat supplied by rain, $Q_G$ is conduction, and $Q_M$ is energy that goes into phase change (i.e. melt or refreezing). Equation 4 holds for a variety of units, provided they are the same for each term; often Watts per meter squared are used, though. If the units of $Q_M$ in Eq. 4 are Watts per meter squared, $Q_M$ is related to $M$ in Eqs. 1-3 through the relationship

$$M = \frac{Q_M \Delta t}{\rho_w L_f} \tag{5}$$

where $\rho_w$ is the density of water ($[\rho_w] = \text{kg m}^{-3}$) and $L_f$ is the latent heat of fusion ($[L_f] = \text{J kg}^{-1}$). In practice, energy balance cryosphere models often solve for melt energy, $Q_M$, as the residual in Eq. 4 (Liston and Elder, 2006a).

Cold content, CC ($[\text{CC}] = \text{m}$), is an alternative means of representing the internal energy state of a snowpack, which is more physically representative than the step functions used in Eqs. 1-3 but simpler than the full energy balance in Eq. 4. CC is defined as

$$\text{CC} = \frac{c_i \rho_s}{L_f \rho_w} d_s \left( T_m - \bar{T}_{s,i} \right) \tag{6}$$

where $c_i$ is the specific heat of ice ($[c_i] = \text{J kg}^{-1} \text{ K}^{-1}$), $\rho_s$ is the density of the snow mass ($[\rho_s] = \text{kg m}^{-3}$), $d_s$ is the depth of snow ($[d_s] = \text{m}$), $T_m$ is the temperature at which melt occurs (i.e. 0 °C; $[T_m] = $ °C), and $\bar{T}_{s,i}$ is the internal temperature of the snow mass (assuming the snow is isothermal or that $T_{s,i}$ is the mass-averaged temperature; $[\bar{T}_{s,i}] = $ °C). CC is simply a statement of the magnitude of sensible energy deficit of the snow mass in terms of the latent energy required to melt the same mass of water. Thus, positive CC values correspond to conditions where more sensible heat transfer is needed before melt can occur and CC equal to zero means that all additional energy inputs will go into latent heat transfer (i.e. will cause melt to occur).

CC is a natural conceptual formulation of the snowpack internal energy state to combine with an SDI or ETI heat flux representation (e.g. Eqs. 1-3) because both are formulated in terms of depth of water equivalent. Hock (2005) states that the

concepts of energy deficit (referring to Van de Wal and Russell, 1994) and CC, or 'negative melt', (referring to Braun and Aellen, 1990) have been incorporated into previous conceptual cryosphere models; however, the above mentioned articles that Hock (2005) cites do not mention their treatment of internal energy. Therefore, we are not aware of any previous distributed cryosphere computational models that incorporate internal energy or 'negative melt' into their formulation using CC.

The surface energy balance, Eq. 4, applies to ice or snowpack. For a complete representation of a multi-layered cryosphere system (either multiple snow layers or snow and ice layers), the internal energy of each layer and heat transfer between the layers must be considered. In many computational models, e.g. that described in Hock and Holmgren (2005), SnowModel (Liston and Elder, 2006a), and the Better Assessment Science Integrating point and Nonpoint Sources modeling framework (BASINS; Brown et al., 2014), sensible and latent energy of snow are considered but sensible energy of glaciers is ignored.

Some models, including SnowModel (Liston and Elder, 2006a), have the option to calculate the state properties of multiple snow layers. A few models, including the COupled Snowpack and Ice surface energy and MAss balance model (COSIMA; Huintjes et al., 2015), calculate the sensible and latent energy of both snow and ice.

## 1.3   Climates of Gulkana and Wolverine Study Domains

Gulkana and Wolverine glaciers are both US Geological Survey (USGS) long-term benchmark glaciers located in Alaska.

Wolverine glacier is on the Kenai Peninsula in relatively close proximity to the Gulf of Alaska while Gulkana is in the Alaska Range and much further from a large body of water (Fig. 1). The model domain we use for Wolverine is larger than that for Gulkana, although the catchment area upstream of the Gulkana streamgage is larger (Table 1). Of note, the elevation ranges within the Wolverine and Gulkana model domains are 1,488 m and 1,129 m, respectively, with the mean elevation of Gulkana being significantly higher than that for Wolverine. Additionally, the glacier coverage within the catchment for Gulkana is 62.3%

and for Wolverine it is 92.9%.

**Table 1.** Select geographic properties of Gulkana and Wolverine study domains and watersheds (Fig. 1). 'Total' refers to properties aggregated over the study domain, 'drainage' refers to properties aggregated over the area contributing to the USGS streamgage, and all elevations are with respect to the study domain.

| Geographic Property | Gulkana | Wolverine |
|---|---|---|
| Total Area, Model ($km^2$) | 52.5 | 74.6 |
| Drainage Area - Model ($km^2$) | 32.0 | 24.2 |
| Drainage Area - USGS ($km^2$) | 31.3 | 24.4 |
| Total Glaciation (%) | 45.0 | 52.6 |
| Drainage Glaciation (%) | 62.3 | 92.9 |
| Mean Elevation (m) | 1661 | 956 |
| Min Elevation (m) | 1132 | 113 |
| Max Elevation (m) | 2261 | 1601 |

As expected from their respective geographies, the climates of Gulkana and Wolverine watersheds are markedly different (Figs. 2 and A1). The hottest air temperatures (i.e. summer temperatures) are similar between Gulkana and Wolverine but Gulkana is significantly colder at the low end of the distribution (i.e. corresponding to colder winter temperatures at Gulkana). At all percentiles in the distribution, Wolverine's precipitation is greater than Gulkana's. Based on Gulkana being relatively drier and colder and Wolverine being wetter and warmer, the watersheds can be classified respectively as continental and maritime (Armstrong and Armstrong, 1987; O'Neel et al., 2014). These climate classifications are consistent with findings for the Alaska region presented in Bieniek et al. (2012) (using cluster analysis of station records) and Simpson et al. (2002) (based on Parameter-elevation Regressions on Independent Slopes Model interpolated climate surfaces; Daly et al., 2002).

Snowpacks in continental climates tend to be much more polythermal and less dense than snowpacks in maritime climates (Armstrong and Armstrong, 1987; DeWalle and Rango, 2008; Ukraintseva, 2009). Higher densities in maritime snowpacks are partially caused by atmospheric conditions during snowfall and also by repeat freezing and melting cycles, which are more common to under maritime conditions. These climatic differences in turn affect the thermal properties of snow and ice because the thermal conductivity of snow is primarily a function of density and water content, and the snowpack properties impact the surface heat flux of glaciers when and where they are snow covered (Sturm et al., 1997; DeWalle and Rango, 2008).

O'Neel et al. (2014) find that Gulkana's glacier mass balance is primarily a function of summer temperatures and Wolverine glacier's mass balance is a function of both summer temperatures and winter precipitation. Interestingly, annual glacier mass balances at Wolverine is much more variable year-to-year than those for Gulkana (O'Neel et al., 2014) . Additionally, the streamflow at Wolverine is less strongly coupled to the summer mass balance compared to Gulkana glacier, even though the fractional glacier coverage at Wolverine is greater, which is consistent with the two glaciers respective classifications as maritime and continental.

## 2   The Conceptual Cryosphere Hydrology Framework (CCHF)

CCHF models alpine and glacier systems using nine process modules (see Figure 3). While it is common for conceptual cryosphere models to combine heat and snow/ice melt into a single formulation (e.g. Eqs. 1-3), CCHF represents these processes using two distinct modules, namely "cryosphere heat transfer" (referred to as the heat module for brevity; described in Section 2.1) and "cryosphere mass flux" (referred to as the mass flux module; described in Section 2.2). While we have programmed multiple representations for each process module in CCHF, we only vary the heat module and mass flux module in this work. We choose this focus because the processes represented in these modules are the most sensitive to climatic differences (resulting from either interregional variability or long-term climatic changes). Section 2 describes the capabilities of CCHF and Sec. 3 outlines the specific methods used to implement CCHF for the present model evaluation.

CCHF is designed to be easy to apply and extend. Presently CCHF can be applied to hourly, daily, or monthly time steps and any spatial grid expressed in uniformly spaced geographic coordinates. We have built in capability for CCHF to read precipitation, mean temperature, minimum temperature, and maximum temperature climate inputs. It would be straightforward, though, to expand CCHF for use with additional climate inputs. CCHF does not include methods for correcting any of the

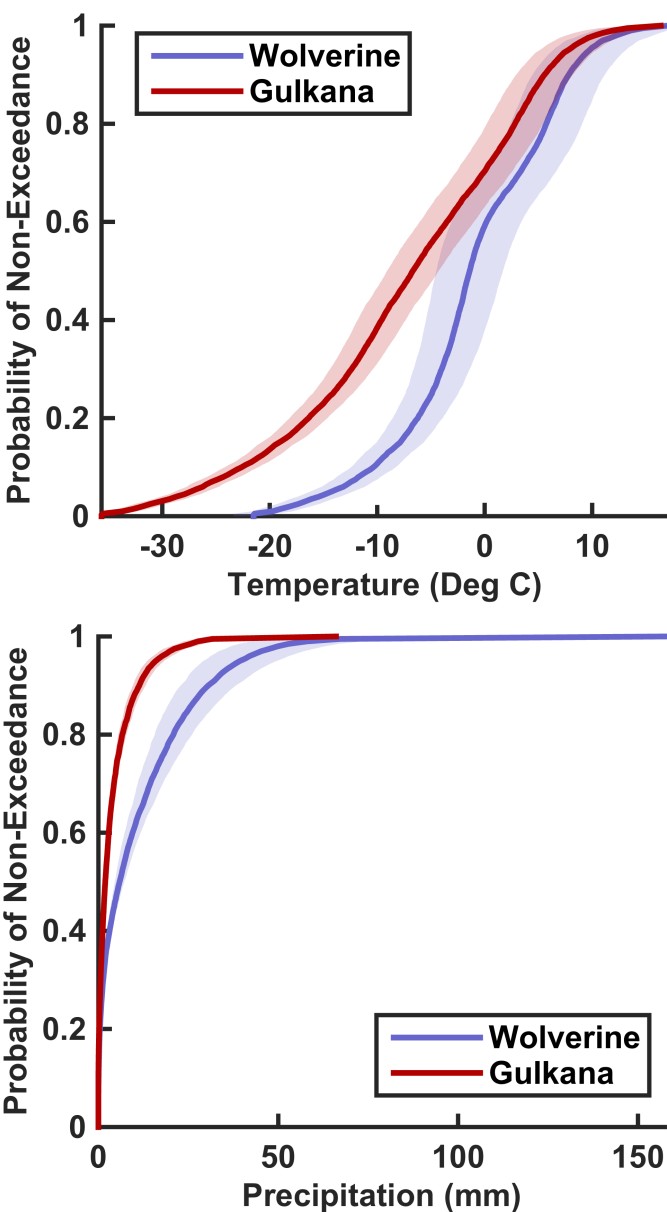

**Figure 2.** Cumulative probability distributions of daily precipitation and mean surface air temperature for the Gulkana and Wolverine watersheds (Fig. 1) from 1980-2009. The lines correspond to the spatially averaged precipitation and temperature values and the shaded region indicates one standard deviation in the spatial distribution of precipitation and temperature within the watershed. Climate representation is derived from the daily downscaled Climate System Forecast Reanalysis (Saha et al., 2010) product discussed in Section 3.1.

climate inputs - any corrections must be undertaken during pre-processing of the climate inputs. All heat flux terms, either conceptual (e.g. degree day factors) or physical (e.g. shortwave radiation) are parameterized within CCHF.

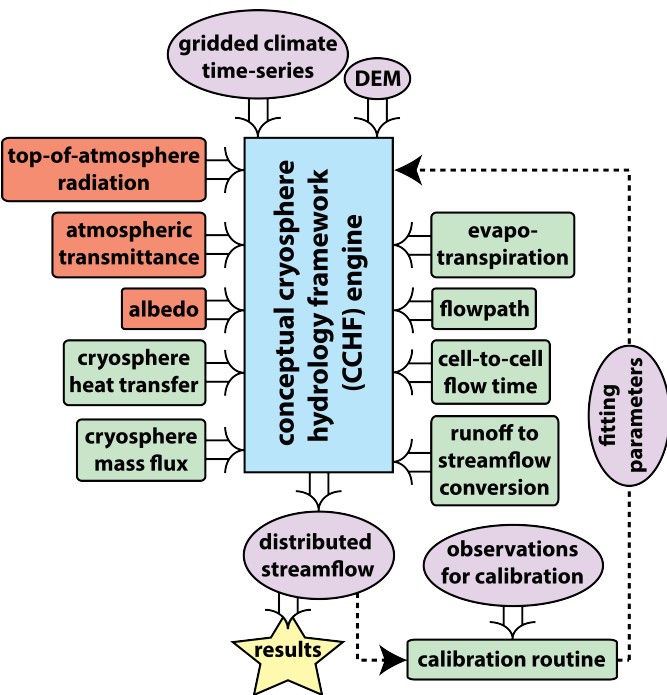

**Figure 3.** Categories of process modules included in the current version of the conceptual cryosphere hydrology framework (CCHF). Rounded blocks refer to process modules and ovals refer to data inputs and outputs. Blocks tinted green denote modules used for all models and blocks tinted orange denote modules only used for models that include shortwave radiation.

The process representations for the seven fixed modules, provided in Table 2, are used because preliminary analysis indicated these representations work well for the conditions under which the CCHF is currently implemented. Section 2.4 discusses the modules involved in translating snow and ice melt and rain into streamflow, which are represented by the four modules on the right hand side of the engine in Fig. 3. In addition to the nine process modules, CCHF also includes a built-in calibration routine (Fig. 3), which we discuss in Section 2.5.

We model the ratio of precipitation that falls as snow, $r_p$ (also referred to as precipitation partitioning), using a linear ramp formulation

$$
r_p = \begin{cases}
1 & \text{if } T_a \leq T_{p,s} \\
\frac{T_{p,r} - T_a}{T_{p,r} - T_{p,s}} & \text{if } T_{p,s} \leq T_a \leq T_{p,r} \\
0 & \text{if } T_{p,r} \leq T_a
\end{cases}
\tag{7}
$$

where $T_{p,s}$ is the threshold air temperature at which all precipitation falls as snow ($[T_{p,s}] = {}^\circ$C) and $T_{p,r}$ is the threshold air temperature at which all precipitation falls as rain ($[T_{p,r}] = {}^\circ$C). The threshold temperatures at which all precipitation falls as snow and rain can be optimized or set. Throughout this work we set $T_{p,s} = 0\ ^\circ$C and $T_{p,r} = 2\ ^\circ$C. We choose these threshold air temperatures because Beamer et al. (2016) optimizes this precipitation partitioning function (Eq. 7) for the Gulf of Alaska

**Table 2.** Process representations for modules that are fixed in the present CCHF implementation. $c_{atm1}$ and $c_{atm2}$ are fitting parameters (unitless), $\Delta T$ is maximum temperature minus minimum temperature ($[\Delta T] = $ K), and $\sum T_{max}$ is the cumulative sum of maximum daily temperatures since the last precipitation event ($[\sum T_{max}] = $ K). Albedo of exposed ice is assumed to be 0.35.

| Module Name | Representation | Source |
|---|---|---|
| top-of-atmosphere radiation | function of latitude, slope, aspect, and day of year | DeWalle and Rango, 2008 |
| atmospheric transmittance | $c_{atm1}\left(1 - \exp(-0.01\,c_{atm2}\,\Delta T^{2.4})\right)$ | DeWalle and Rango, 2008 |
| albedo (snow) | $0.90 - 0.55 \log\left(\sum T_{max}\right)$ | Pellicciotti et al., 2005 |
| potential evapotranspiration | Eq. 22 | Allen et al., 1998; Lu et al., 2005 |
| flowpath | Sec. 2.4 | manually determined |
| cell-to-cell flow time | Eq. 23 | Johnstone et al., 1949 |
| runoff to streamflow conversion | Sec. 2.4 | Moore et al., 2012 |

and finds that these temperature thresholds perform better than other sets of threshold temperatures tested. Of note, Beamer et al. (2016) also find that the performance of SnowModel (the model they implement; Liston and Elder, 2006a) is insensitive to modest perturbations in the precipitation partitioning parameters. A likely reason that the model is insensitive to these parameters for this region is that Alaska has strong seasonal variations in temperature and winter temperatures are typically much colder than the mixed-phase precipitation zone. Therefore, we do not expect the precise values to significantly impact results of our study and by setting the values fixed we reduce the dimensionality of our model calibration.

When rain is incident on snow, it add to the liquid water content of the snowpack. From which point, rain on snow is accounted for in the same manner as melted snow (see description in Sec. 2.4). Rain incident directly on the glacier surface is treated in the same manner as when it is incident on bare ground (see Sec. 2.4). An implication is that we do not allow refreezing on the bare glacier surface.

## 2.1 Cryosphere Heat Transfer

Implementing the existing conceptual cryosphere models described by Eqs. 1-3 requires transforming their units because all heat terms in CCHF have units of Watts per meter squared. We therefore define heat representations to use in CCHF that are

analogs to these previous conceptual representations (i.e. Eqs. 1-3, respectively):

$$H_{\text{SDI}} = c_{fm} \left( T_a - T_0 \right) \tag{8}$$

$$H_{\text{ETI(H)}} = \left( c_{hoc,s/i} I + c_{fm} \right) \left( T_a - T_0 \right) \tag{9}$$

$$H_{\text{ETI(P)}} = c_{pel} \left( 1 - \alpha \right) I + c_{fm} \left( T_a - T_0 \right) \tag{10}$$

where the $H$ terms are the net heat fluxes ($[H] = $ W m$^{-2}$; subscript "SDI" refers to the generic simple degree index model, "ETI(H)" refers to the ETI model described in Hock, 1999, and "ETI(P)" refers to the ETI model described in Pellicciotti et al., 2005), $c_{fm}$ is the degree-index factor ($[c_{fm}] = $ W m$^{-2}$ °C$^{-1}$), $c_{hoc,s/i}$ represents two fitting parameters, $c_{hoc,s}$ and $c_{hoc,i}$, which scale incoming shortwave radiation, $I$, incident upon snow and ice, respectively ($[a_{h,s/i}] = $ °C$^{-1}$), and $c_{pel}$ is a fitting parameter to scale the magnitude of shortwave radiation in the $H_{\text{ETI(P)}}$ representation (unitless).

We also implement and assess a novel conceptual heat representation, which we refer to as the Longwave, Shortwave, and Temperature (LST) formulation. The LST representation is similar to ETI(P) except that LST includes a longwave radiation balance term and LST does not have a fitting parameter to scale shortwave radiation. The LST heat flux, $H_{\text{LST}}$ ($[H_{\text{LST}}] = $ W m$^{-2}$) is

$$H_{\text{LST}} = \left( 1 - \alpha \right) I + c_{fm} T_a + \sigma \left( \varepsilon_a \left( T_a + 273.15 \right)^4 - \left( T_{s,s} + 273.15 \right)^4 \right) \tag{11}$$

where $\sigma$ is the Stefan-Boltzmann constant ($\sigma = 5.67 \times 10^{-8}$ W m$^{-2}$ K$^{-4}$), $\varepsilon_a$ is the effective atmospheric emissivity (unitless), and $T_{s,s}$ is the snow surface temperature ($[T_{s,s}] = $ °C). The LST formulation assumes $\varepsilon_a$ to be 0.7 when there is no precipitation and 1 otherwise, which roughly translates to clear-sky and cloudy conditions (Hock, 2005; Sedlar and Hock, 2009). Unlike the SDI, ETI(H), and ETI(P) heat transfer representations (Eqs. 8-10), the LST representation requires modeling the snowpack and ice surface temperatures, which is discussed in Sec. 2.2.

**2.2   Cryosphere Mass Flux**

We implement two process representations within the mass flux module to relate the net heat flux to changes in the internal energy and mass of snow and ice. The first representation is a step function, denoted as $\theta$, which is implicit in the SDI and ETI formulations used in existing conceptual models (i.e. Eqs. 1-3). The second mass flux representation is based on the cold content (Eq. 6), and is denoted as CC. In both the $\theta$ and CC mass flux representations, heat is translated into melt potential, 25   $M_P$ ($[M_P] = $ m), which is analogous to $M$ in Eq. 5, through the relationship

$$M_P = \frac{H \, \Delta t}{\rho_w \, L_f} \tag{12}$$

Melt in the $\theta$ mass flux representation, $M_\theta$ ($[M_\theta] = $ m), is equal to the melt potential (Eq. 12) when air temperature is greater than a threshold value, which is described mathematically as

$$M_\theta = \begin{cases} M_P & \text{if } T_a > c_{thr} \\ 0 & \text{otherwise} \end{cases} \tag{13}$$

where $c_{thr}$ is taken to be a fitting parameter ($[c_{thr}] = {}^\circ$C). Therefore, combining the respective heat flux representations $H_{\mathrm{SDI}}$, $H_{\mathrm{ETI(H)}}$, and $H_{\mathrm{ETI(P)}}$ with the $\theta$ mass flux representation essentially allows us to reproduce the previous conceptual models presented in Eqs. 1-3, albeit with different units for many of the fitting parameters.

Cold content in CC mass flux representation, is denoted as $w_c$ ($[w_c] = $ m) to distinguish it from the general definition of CC in Eq. 6. $w_c$ is calculated as

$$w_{c,i} = \begin{cases} w_{c,i-1} - c_{cc} M_{P,i} & \text{if } M_{P,i} < 0 \\ w_{c,i-1} - M_{P,i} & \text{otherwise} \end{cases} \tag{14}$$

where subscript $i$ refers to the current time step, subscript "$i-1$" refers to the previous time step, and $c_{cc}$ is a unitless fitting parameter that provides a hysteresis in the accumulation and depletion of $w_c$, and $M_P$ can be either positive or negative (determined by sign of $H$ in Eq. 12). The lower bound on $w_c$ is zero (i.e. $w_c$ is set to zero if it becomes negative) and $w_c$ is always zero at grid locations without snowpack, including those with ice but no snow. Physically, the CC hysteresis relates to differences in the conduction of energy through the snowpack during accumulation (dry) versus ablation (wet) conditions. Melt in the CC mass flux representation, $M_{\mathrm{CC}}$ ($[M_{\mathrm{CC}}] = $ m), is calculated as

$$M_{\mathrm{CC}} = \begin{cases} M_P - w_c & \text{if } M_P > w_c \\ 0 & \text{otherwise} \end{cases} \tag{15}$$

While snowpack temperature cannot be modeled in the $\theta$ representation, the CC representation models the average internal snowpack temperature, $\bar{T}_{s,i}$, as

$$\bar{T}_{s,i} = -c_{tsn} w_c \tag{16}$$

where $c_{tsn}$ is a fitting parameter to translate $w_c$ into internal temperature ($[c_{tsn}] = {}^\circ$C m$^{-1}$). Thus, the assumption is made in the CC representation that the snowpack is isothermal (i.e. $\bar{T}_{s,i} = T_{s,s}$). When liquid water in the snowpack is greater than 0.5% of solid snow water equivalent (SWE), the snowpack temperature is set to $0\ {}^\circ C$. The LST heat representation is the only formulation implemented here that uses Eq. 16 and in this case it is assumed that the snow surface temperature, $T_{s,s}$ in Eq. 11, is equal to the snowpack's average internal temperature, $\bar{T}_{s,i}$ in Eq. 16. This is equivalent to a strict isothermal snowpack assumption, which is more realistic for wet and warm snowpacks than for dry and cold snowpacks. Based on the climates of Gulkana and Wolverine (Fig. 2 and A1), we expect this to be a better assumption for Wolverine than Gulkana.

Both the $\theta$ and CC mass flux representations contain a maximum snowpack liquid water content, MLC ($[\mathrm{MLC}] = $ m), represented by

$$\mathrm{MLC} = c_{slq} \mathrm{SWE}_s \tag{17}$$

where $c_{slq}$ is a unitless fitting parameter and SWE$_s$ is the snowpack's solid water content expressed as depth of liquid water equivalent ($[\mathrm{SWE}_s] = $ m). All liquid water in excess of the snow's liquid water holding capacity drains from the snowpack

during each time step. In the CC formulation, the snowpack's liquid water is refrozen when CC is greater than 0, and the CC is reduced in direct proportion to the depth of water frozen. Only water that drains from the snowpack is counted as a change in the snowpack's total water content.

## 2.3  Glacier Treatment

Fully representing ice processes requires accounting for the ice's internal energy. Conceptual models typically do not do this, instead assuming the the ice temperature is 0 °C when there is no snow present and the heat flux into the ice is positive (Hock, 1998, 1999). The present version of CCHF does not include any mass flux modules that account for the internal energy of ice, although this is an area that we believe should be developed in the future. Instead, CCHF assumes the glacier surface albedo is 0.35 and that the glacier's surface temperature is the minimum of the air temperature in the previous time step and 0 °C. The glacier's surface temperature only impacts net longwave heat flux calculated in the LST model.

During a given time step, ice melt potential, $M_{P,ice}$ ($[M_{P,ice}] = $ m), is calculated as the remainder of potential melt after snow melt is calculated, i.e.

$$
M_{P,ice} = \begin{cases} M_P - M & \text{if } M_P > M \\ 0 & \text{otherwise} \end{cases} \tag{18}
$$

Actual ice melt, $M_{ice}$ ($[M_{ice}] = $ m), is scaled by a unitless fitting parameter, $c_{glc}$, that accounts for energy differences in melting ice versus snow (e.g. accounting for differences in heat conduction between ice and snow). Thus, ice melt is calculated as

$$
M_{ice} = c_{glc} \frac{H_{\text{ice}}}{H} M_{P,ice} \tag{19}
$$

where $H_{\text{ice}}$ is the heat flux for the ice surface properties and $H$ is the equivalent heat flux for snow surface properties. For a given model implementation, both $H$ and $H_{\text{ice}}$ correspond to the same heat flux representation (i.e. one representation from Eqs. 8-11). For example, for the model where $H_{ETI(P)}$ is used as the heat representation, the only difference between $H$ and $H_{ice}$ in Eq. 19 is due to the difference in surface albedo. The purpose of Eq. 19 is that it allows ice melt to occur in the same time step in which the snowpack has completely melted from a grid cell. Eq. 19, which is essentially a ratio to scale the melt potential available for ice, approximates the portion of energy that is still available for ice melt after all snow has melted during the current time step. Over short time steps, the error incurred by not allowing snow and ice melt to occur for a given grid cell during the same time step is small; however, including Eq. 19 allows the model to scale better between small and large time steps.

## 2.4  Melt and Rain to Streamflow

Once liquid water drains from the snowpack, ice melt occurs, or rain is incident on a grid cell without snow, CCHF models its transport through the landscape with the four modules on the right side of Fig. 3. Two general phases of this transport are (1) transition from snow or ice melt to runoff and (2) flow through the stream network. To model phase one, we implement a leaky groundwater bucket model as described in Moore et al. (2012). In the bucket representation, all liquid water released from

snow, snlr ([snlr] = m), liquid water released from ice, iclr ([iclr] = m), and rain incident on a grid cell without snow ([rain] = m) enters the groundwater bucket. The water in the bucket is described as the soil moisture, SM ([SM] = m), with a total soil moisture capacity, $c_{smc}$ ([$c_{smc}$] = m; $c_{smc}$ is a fitting parameter). SM is updated during each time step as

$$
\text{SM}_i = \begin{cases} \text{SM}_{i-1} + \text{rain}_i + \text{snlr}_i + \text{iclr}_i - r_{f,i} - \text{PET}_i & \text{if no snow or ice present} \\ \text{SM}_{i-1} + \text{rain}_i + \text{snlr}_i + \text{iclr}_i - r_{f,i} & \text{otherwise} \end{cases} \tag{20}
$$

where subscript $i$ refers to the value during the $i^{\text{th}}$ time step and PET is potential evapotranspiration ([PET] = m; described below). Runoff, $r_f$ ([$r_f$] = m$^3$ days$^{-1}$), in the bucket model is then calculated based on SM and $c_{smc}$ as

$$
r_f = \begin{cases} A\left((1 + c_{dr})\,\text{SM} - c_{smc}\right) & \text{if SM} \geq c_{smc} \\ A\,c_{dr}\,\text{SM} & \text{otherwise} \end{cases} \tag{21}
$$

where $A$ is the grid cell's area ([$A$] = m$^2$) and $c_{dr}$ is the drain rate of soil moisture from the leaky groundwater bucket per day ([$c_{dr}$] = days$^{-1}$; fitting parameter ranging from zero to one).

PET is calculated using a formulation of the Hamon Equation (Hamon, 1961) based on Allen et al. (1998) and Lu et al. (2005), in which

$$
\text{PET} = 218.39 \times c_{pet} \times N \left( \frac{6.108 \exp\left[ \frac{17.27\,T_a}{T_a + 273.15} \right]}{T_a + 273.15} \right) \tag{22}
$$

where $c_{pet}$ is a unitless fitting parameter and $N$ is the number of daylight hours ([$N$] = hrs), which is calculated according to the formulation presented in DeWalle and Rango (2008). PET is set to zero when snow or ice is present.

Flow direction between grid cells in CCHF can be input by the user (in the form of an Environmental Systems Research Institute - ESRI - formatted flow direction grid) or calculated automatically using algorithms from the TopoToolbox (Schwanghart and Kuhn, 2010). The travel time through each cell, $t_t$ ([$t_t$] = s) is then calculated using a formulation by Johnstone et al. (1949),

$$
t_t = 3600\, c_{tvl} \sqrt{ \frac{\Delta l}{\sqrt{m}} } \tag{23}
$$

where $c_{tvl}$ is a unitless fitting parameter, $\Delta l$ is the distance between the centroids of grid cells in the flowpath ([$\Delta l$] = m), and $m$ is the slope between cells in the flowpath expressed as a ratio (unitless).

Streamflow is then routed downhill along the flowpath determined by the flow direction grid using either the "lumped" or Muskingum methods (Bedient et al., 2012). The lumped method does not allow dispersion of streamflow, whereas the Muskingum method does allow dispersion according to the formulation presented in Bedient et al. (2012). The results of both

flow routing methods are comparable for simulating small watersheds at large time steps; however, dispersion becomes more important as the size of the watershed increases relative to the time step (Bedient et al., 2012). In this work we use the lumped method since the model domains are small.

## 2.5 Calibration Routine

Several calibration routine options are built into the CCHF, including a genetic algorithm (Wang, 1991), Monte Carlo simulation (Sawilowsky and Fahoome, 2003), Particle Swarm Optimization (PSO; Poli et al., 2007), and a hybrid algorithm developed by us. The hybrid algorithm works by implementing the Monte Carlo routine for the first several calibration generations (ranging from four to 15 generations based on the number of fitting parameters) and updating the fitting parameters in the following generations using PSO until two consecutive generations stagnate to the same fitness score (referred to here as an initial stagnation). Once an initial stagnation occurs, the hybrid algorithm uses a combination of Monte Carlo simulations and linear sensitivity analysis to alternately add diversity to the population (Monte Carlo simulation) and explore the local parameter space (linear sensitivity). If a new local optima is found, PSO is reinitiated. The process repeats until the same best fitness score is returned for 15 consecutive generations (referred to here as terminal stagnation). A population of 30 parameter sets is used in each generation of the optimization process.

Calibration in the CCHF can be conducted in one or multiple stages. If multiple stages are implemented, the user must determine which categories of fitting parameters to optimize in each stage. The three performance metrics we use are the Nash Sutcliffe Efficiency (NSE; Nash and Sutcliffe, 1970) score, the Kling Gupta Efficiency (KGE; Gupta et al., 2009) score, and the the Moderate Resolution Imagining Spectroradiometer (MODIS; Hall et al., 2006) SCA comparison paradigm presented in Parajka and Blöschl (2008), which we refer to as the Parajka and Blöschl Error (PBE; Parajka and Blöschl, 2008). The NSE and KGE scores are computed for glacier stake and streamflow comparisons while the PBE score is computed to compare SCA observations to modeled snow as described below.

The NSE score is extensively used to assess hydrologic models and, as is shown in Gupta et al. (2009), can be decomposed into the form

$$\text{NSE} = 2\alpha r - \alpha^2 - \beta^2 \tag{24}$$

where $\alpha$ is the ratio of the model standard deviation to the observed standard deviation, $r$ is the linear correlation coefficient, and $\beta$ is the bias normalized by the observed standard deviation. From Eq. 24 it is evident that the NSE score unequally weights standard deviation, correlation coefficient, and bias. Therefore, Gupta et al. (2009) propose using the KGE score, defined as

$$\text{KGE} = 1 - \sqrt{(r-1)^2 + (\alpha-1)^2 + (\beta-1)^2} \tag{25}$$

which equally weights errors in the correlation, standard deviation, and the non-dimensional bias.

A perfect KGE or NSE score is one and the negative bound on the scores is negative infinity. An NSE score of zero indicates the mean of the observation time-series is as as good of a predictor as the modeled time-series. Note though, that a KGE score of zero does not share this interpretation. The advantage of using the KGE for calibration instead of the NSE is that the NSE is sensitive to errors in extreme values and less sensitive to errors in the overall distribution relative to the KGE metric (Legates and McCabe Jr, 1999).

Using MODIS SCA observations for model evaluation presents a unique challenge since MODIS observes snow and ice cover while CCHF models SWE. The PBE score overcomes this issue by calculating two error metrics - snow overestimation

error, $S_E^O$ (unitless), and snow underestimation error, $S_E^U$ (unitless) - which respectively capture instances where the model grid cell contains snow but the MODIS pixel does not report SCA and instances where the model grid cell does not contain snow but the MODIS pixel reports SWE. $S_E^O$ and $S_E^U$ are calculated as

$$S_E^O = \frac{1}{ml} \sum_{j=1}^{l} m_o \wedge (\text{SWE} > \xi_{\textbf{SWE}}) \wedge (\text{SCA} = 0) \tag{26}$$

$$S_E^U = \frac{1}{ml} \sum_{j=1}^{l} m_u \wedge (\text{SWE} = 0) \wedge (\text{SCA} > \xi_{\textbf{SCA}}) \tag{27}$$

where $m$ is the number of number of MODIS time steps in which less than 60% of the MODIS image over the entire domain is cloud covered (images with cloud cover greater than this threshold are not used), $l$ is the number of grid cells that are not glaciated or have permanent snow cover (these MODIS pixels are also removed from analysis as explained below), the summation over $j$ loops over all MODIS pixels that contribute to $l$, $m_o$ is the number of MODIS time steps for which MODIS reports 0% snow cover at the current grid cell and modeled SWE is greater than a threshold value set by $\xi_{\textbf{SWE}}$ ($[\xi_{\textbf{SWE}}] =$ m), and $m_u$ is the number of MODIS time steps where no SWE is modeled at the current grid cell but MODIS observes snow cover greater than $\xi_{\textbf{SCA}}$ (unitless). Due to potential issues in which MODIS incorrectly classifies permanent ice as snow, we remove all MODIS pixels from analysis when a given pixel is classified as snow during more than 90% of the time steps. Thus, MODIS data are only used in model evaluation for grid cells with seasonal snow cover and not for grid cells that MODIS classifies as having permanent snow.

The PBE score is calculated from $S_E^O$ and $S_E^U$ using the weighting function

$$\text{PBE} = w_1 S_E^O + w_2 S_E^U \tag{28}$$

where PBE is the snow cover error used in model assessment (unitless; range is 0 to 1, where 0 indicates no error) and $w_1$ and $w_2$ are unitless weighting factors used to scale the model overestimation and underestimation errors, respectively.

$S_E^O$ and $S_E^U$ respectively decrease as $\xi_{\textbf{SWE}}$ and $\xi_{\textbf{SCA}}$ increase. Thus, errors are largest with $\xi_{\textbf{SWE}} = 0$ m and $\xi_{\textbf{SCA}} = 0\%$. Parajka and Blöschl (2008) set the values of $\xi_{\textbf{SWE}}$ and $\xi_{\textbf{SCA}}$ in order to balance $S_E^O$ and $S_E^U$. Unfortunately it is not possible to balance $S_E^O$ and $S_E^U$ apriori in the present implementations because the sensitivity of $S_E^O$ and $S_E^U$ with respect to $\xi_{\textbf{SWE}}$ and $\xi_{\textbf{SCA}}$ changes between model domains, model formulations, and with different fitting parameter values. We set $\xi_{\textbf{SWE}}$ to 10 mm and $\xi_{\textbf{SCA}}$ to 10% in order to increase the sensitivity of the calibration to both $S_E^O$ and $S_E^U$. We do not set the $\xi$ values to 0 in order to recognize that there can be classification errors in MODIS SCA observations for pixels with low snow cover and because CCHF assumes all snow is uniformly distributed over the grid cell, whereas at very low SWE values this is not likely to be accurate. We set $w_1$ and $w_2$ in Eq. 28 to 5 because we find through initial calibration analysis that this tends to result in PBE having similar magnitudes to KGE values computed for glacier stake observations.

# 3 Methods for Model Comparison

In this work we implement seven models, shown in Table 3, which are combinations of the heat module and mass flux module representations described in Secs. 2.1 and 2.2, respectively. Our primary objectives in developing the model evaluation strategy are to reduce equifinality and identify differences in accuracy and precision between the models. We reduce equifinality through utilizing a multi-objective evaluation criteria comprised of observations from multiple glacier stakes, MODIS SCA images, and streamgage data, calibrating the models in two stages, and conducting calibration for ten consecutive water years in which all observation data are available. Reducing equifinality helps enable intercomparison of model accuracy and precision. We additionally assess how much equifinality is present in each of the seven calibrated models. Section 3.1 describes the inputs used in these implementations of the model, Sec. 3.2 then provides information on the observation data used, and Sec. 3.3 explains the evaluation strategy.

**Table 3.** Heat module and mass flux module combinations that we assess, described respectively in Secs. 2.1 and 2.2. In all cases the seven fixed module formulations are those presented in Table 2.

| Model Label | Heat Eq. # | mass flux Eq. # |
|---|---|---|
| SDI-$\theta$ | 1 | 13 |
| SDI-CC | 1 | 14 |
| ETI(H)-$\theta$ | 2 | 13 |
| ETI(H)-CC | 2 | 14 |
| ETI(P)-$\theta$ | 3 | 13 |
| ETI(P)-CC | 3 | 14 |
| LST-CC | 11 | 14-16 |

## 3.1 Model Inputs

CCHF presently supports inputs of precipitation and temperature (mean, minimum, and maximum) time-series, a digital elevation model (DEM; we use the WorldClim 30 arcsecond DEM, Hijmans, 2011), a data file containing outlines of glacier or ice cover, and a flow direction raster (FDR). The FDR can be generated automatically within CCHF using algorithms from the TopoToolbox, Schwanghart and Kuhn, 2010); however, we manually delineate the watersheds and create the FDR using a combination of ESRI algorithms and visual analysis.

The climate time-series inputs used here are derived from the Climate Forecast System Reanalysis (CFSR), which is produced by the National Centers for Environmental Prediction (NCEP) (Saha et al., 2010). CFSR is an hourly climate product available from 1979-2010 as 0.312 degree grids. Wang et al. (2011) and Lader (2014) find that CFSR represents precipitation and temperature variability as well or better than other reanalysis products such as MERRA (Rienecker et al., 2011) and ERA-Interim (Dee et al., 2011) for the Alaska region. Further, Beamer et al. (2016) find that CFSR performs well relative to other

reanalysis products for Gulkana and Wolverine glaciers and also better reproduces total volumes of water flux into the Gulf of Alaska.

We temporally aggregate the hourly 0.312 degree CFSR product to daily time steps. Precipitation and mean temperature are used in all instances, while minimum and maximum temperature are only used in model formulations that include shortwave radiation. We input the temporally aggregated 0.312 degree CFSR product into MicroMet (Liston and Elder, 2006b). Micromet resamples the input time-series to the spatial grid of the DEM and applies temperature lapse rates and precipitation correction factors based on elevation distribution, which vary as a function of the day of the year, creating a 30 arcsecond daily time-series for each climate variable.

Glacier outlines in CCHF can be supplied in gridded format using the same grid as the input DEM or can be input as a shapefile. In the case where a shapefile is input to the model, CCHF determines the fractional area of glacier coverage within each of the model's grid cells. We use the Randolph Glacier Inventory version 5 shapefiles (Pfeffer et al., 2014) to generate the glacier outlines in this work and hold these outlines fixed over the duration of each model run.

## 3.2 Observation Data

The three types of observation data we use to assess model performance are USGS streamgage measurements of flowrate (referred to in the results as "flow"), Moderate Resolution Imagining Spectroradiometer (MODIS; Hall et al., 2006) images of SCA (referred to as "SCA"), and USGS glacier stake measurements of changes in snow and ice water equivalent (referred to as "stake"). Gulkana and Wolverine are both considered long-term benchmark glaciers by the USGS, with the implication that glacier stakes and streamgages have been present at both locations for multiple decades, albeit with some interruptions as described below.

The USGS has been measuring glacier and snow accumulation and ablation at three locations on Gulkana glacier from 1974 through the present (with an additional stake from 1990 through the present) and three locations on Wolverine from 1965 through the present (Van Beusekom et al., 2010). Glacier stakes measure changes in depth of snow and ice at one location between two observation dates. These changes in depth are then converted to changes in mass through measuring the density of snow and assuming a density for ice. Wolverine glacier is entirely free of surface debris, while a portion of the ablation zone of Gulkana glacier contains scattered debris cover, but we model both glaciers as debris-free. In our analysis, we treat each glacier stake observation equally, regardless of duration, season, or magnitude of change.

There are streamgages located slightly below the termini of Gulkana and Wolverine glaciers. The streamgages were installed in 1966 and are maintained by the USGS. The temporal coverage of the streamgages is not continuous. For example, the streamgage at Gulkana is missing measurements from October 1978 through September 1989 and the streamgage at Wolverine does not have measurements from October 1978 through September 2000. The USGS also cautions that there may be significant errors in the flows, especially at high flows, because the streamgages are located in geomorphologically active channels.

We use the MODIS MOD10A2 version 5 SCA estimates, which are 8-day maximum SCA estimates projected to 500 m using a Sinusoidal tile grid, available from March 2000 through the present (Hall et al., 2006). We use the 8-day images rather than daily because cloud cover obstructions are a significant hindrance in the daily product and Zhou et al. (2005) achieve

better performance using the 8-day time-series rather than the daily. MOD10A2 version 5 provides a binary classification for each pixel (i.e. either snow, no snow, or no decision). When 60% or more of the pixels in a current image corresponding to one of the study domains is no decision, we set the entire image to no decision (this threshold is also used in Parajka and Blöschl, 2008). We then reproject the image to geographic coordinates and resample the images to our model domains using bilinear

interpolation. Therefore, the SCA values used in model assessment can range from 0-100%.

## 3.3  Evaluation Strategy

Our primary goal with the model intercomparison is understanding how each of the seven models in Table 3 performs under different climatic regimes (as a partial analog for climate change applications) and across geographies (because models are often calibrated for gaged watersheds but applied to non-gaged watersheds). Due to this, we calibrate each model to both

Gulkana and Wolverine domains separately and then validate each model for the opposite watershed (i.e. we perform a total of 14 model calibrations and 14 model validations). We spin up each of the model runs from September 1997 through August 2000 and then conduct each calibration or validation assessment from September 2000 through August 2010.

The combined use of MODIS, glacier stake, and streamgage observations has been shown to significantly improve model performance and better differentiate performance of model parameter set ensemble members (Konz and Seibert, 2010; Finger

et al., 2011). By only evaluating models for the period 2000-2010, we are able to use all three observation types and therefore utilize a stronger evaluation criteria for each model. We view the ten year calibration and validation periods as necessary to assess performance over a range of climatic conditions and reduce equifinality. Often models are assessed for shorter periods (e.g. Singh et al., 2000, Liston and Mernild, 2012, and Hock and Holmgren, 2005) but there is evidence that assessing models over short periods can lead to incorrect assessments (e.g. see Razavi and Tolson, 2013).

We calibrate each model in two stages. In the first stage, parameters relating to cryosphere modules (see Fig. 3) are optimized by maximizing the average of one minus the PBE score for MODIS SCA relative to modeled SWE and the KGE value between measured and modeled changes in SWE and ice water equivalent at glacier stake changes (see details in Sec. 2.5). In the second stage, the calibrated cryosphere parameters are then set to their optimized values and all remaining parameters (i.e. those related to runoff generation and routing processes; see Sec. 2.4) are optimized by maximizing the KGE score for modeled and measured

streamflow at the USGS streamgage locations. Since glacier stake observations are available at multiple points, fitness scores during the first stage of calibration are calculated at each glacier stake location and then averaged to produce a single glacier stake score for each parameter set.

We assess equifinality due to the cryosphere process representations by conducting validation on the 100 parameter sets that perform best during the first stage of calibration. We set the fitting parameters corresponding to non-cryosphere parameters (i.e.

those calibrated in the second stage) to their calibrated values. Thus, we are only investigating equifinality in fitting parameters related to modeling cryosphere processes. We provide equifinality assessment statistics in all tables of validation run results. Specifically, we include (1) validation performance for the parameter set that performs best during calibration, (2) the mean validation performance for the 100 best performing calibration runs, and (3) the standard deviation of validation performance

for the 100 best performing calibration runs. The general format we use for reporting these statistics is "$x_1$; $x_2 \pm x_3$", where the $x_i$ refer to the three numbers statistics from the previous sentence.

## 4   Results and Discussion

Each of the seven models (Table 3) is calibrated and validated using the procedure outlined in Sec. 3.3. The corresponding
PBE scores (for SCA observations) and KGE scores (for stake and flow observations) are summarized in Fig. 4 (full results, including NSE scores and metrics to convey equifinality, are provided in Tables B3, B4, and B5). The corresponding calibrated fitting parameter values are provided in Tables B1 and B2. While Fig. 4 efficiently summarizes the results, trends become more apparent when model performance is aggregated in specific ways. We therefore separate this section into several subsections to highlight interesting aspects of the model intercomparison results. Some of the findings are that (1) which mass flux module
representation performs best depends on the watershed (Sec. 4.1), (2) overall performance varies between the two watersheds (Sec. 4.2), and (3) which model formulations performs best depends on the observation variable considered but the ETI(P)-CC and LST-CC models stand out as the most accurate and robust between regions (Sec. 4.3). Section 4.5 then summarizes general observations.

We also want to note at the outset that the choice of metric, as well as the overall assessment methodology, are significant
determinants of perceived model performance. We primarily use and report the KGE score (Eq. 25) for glacier stake and flow observations and the PBE score (Eq. 28) for SCA observations, although Tables B3, B4, and B5 provide model performance quantified by the NSE score (Eq. 24) as well. As is displayed in Sec. 2.5, the KGE and NSE scores can be decomposed into three principal sources of error (i.e. correlation, standard deviation, and bias). As is outlined in Gupta et al. (2009), the relative importance of each of the three additive terms in Eqs. 24 and 25, $F_i$, can be calculated as

$$20 \quad F_i = \frac{f_i}{\sum\limits_{j=1}^{3} f_j} \tag{29}$$

where $f_i$ are values of the respective terms. To demonstrate how each error component impacts the overall KGE and NSE scores, Table 4 provides the stake error components corresponding to validation runs for the LST-CC model. What is important to note is that the first term of the NSE score ($2 \alpha r / \sum [n_i]$) is negative for both watersheds, and therefore masks some of the error in the other two terms, which relate to the standard deviation and bias. This potential for error masking in the NSE score
explains one factor contributing to higher NSE scores relative to the corresponding KGE scores (see Tables B3, B4, and B5).

Another finding from Table 4 is that the largest sources of KGE error for Gulkana are the correlation coefficient and standard deviation components, while the largest sources of error for Wolverine are the correlation coefficient and bias components. The cause of these respective errors cannot be ascertained from Table 4 without more investigation; however, this type of information may be informative for improving the future model representations.
Many other modeling studies implement similar conceptual models and report "better" model performance. Better is placed in quotation marks because, as is shown in Table 4, what constitutes better depends on the metric used and how the model is

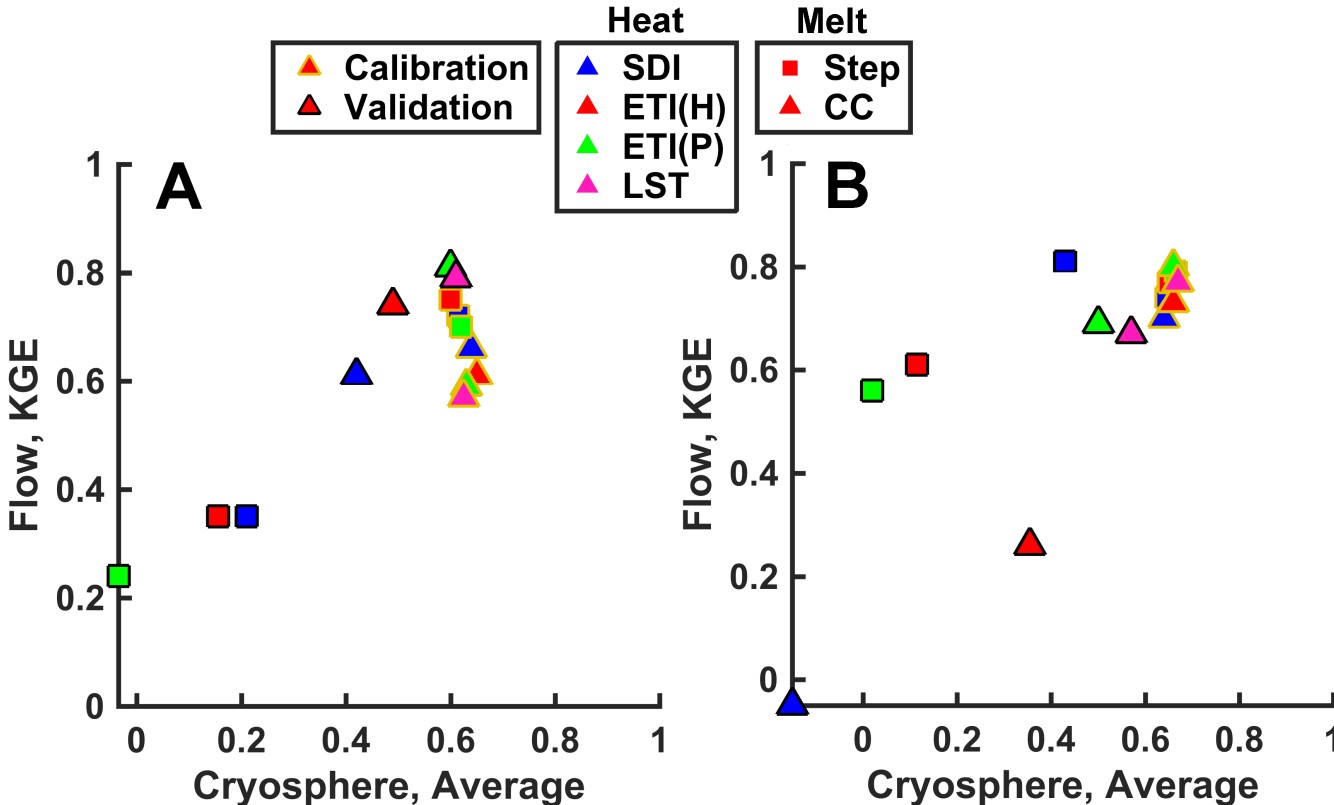

**Figure 4.** Model flow performance during calibration and validation as a function of combined cryosphere performance using the methodology and metrics described in Secs. 2.5 and 3.3. In subplot A, models are calibrated for Gulkana and validated for Wolverine; in subplot B, models are calibrated for Wolverine and validated for Gulkana. As indicated in the plot legends, the marker border indicates phase of assessment, face color indicates the heat module representation, and shape indicates the mass flux module representation.

assessed. Commonly glacier models are evaluated for only a few years (e.g. Liston and Mernild, 2012) or only during the melt season (e.g. Singh et al., 2000; Pellicciotti et al., 2005). Additionally, models evaluated using fewer types of observations tend to appear to perform better (Konz et al., 2010; Finger et al., 2015). It has also been demonstrated by Razavi and Tolson (2013) that when calibrating and validating hydrologic models, use of short durations can lead to inaccurate assessments of model performance and increase model equifinality. In the context of identifying robust models for use in projecting the impacts of climate change, it is therefore necessary to ensure that the validation utilizes a sufficient number of types of observation data, that the evaluation period is sufficiently long, and that evaluation is conducted for multiple climatic regimes.

### 4.1 Mass Flux Module

Average performance of the two mass flux representations, $\theta$ and CC, tends to vary by watershed (Table 5). On average the $\theta$ representation performs better for Gulkana and the CC representation performs better for Wolverine. A difference between

**Table 4.** Sources of stake measurements errors in the Kling Gupta Efficiency (Eq. 25) score in the LST-CC model during validation for Gulkana and Wolverine glaciers. The three error types are the linear correlation coefficient, $r$, the ratio of the model standard deviation to the observed standard deviation, $\alpha$, and the bias normalized by the observed standard deviation, $\beta$. $\sum[k_j]$ and $\sum[n_j]$ refer to the sum of additive terms for the KGE score and NSE score, respectively (i.e. the denominator on the righthand side of Eq. 29.

| Error Term | Gulkana | Wolverine |
|---|---|---|
| $\lvert r - 1 \rvert$ | 0.83 | 0.96 |
| $\lvert \alpha - 1 \rvert$ | 0.82 | 0.73 |
| $\lvert \beta - 1 \rvert$ | 0.59 | 0.83 |
| $(r-1)^2 / \sum[k_j]$ | 0.40 | 0.43 |
| $(\alpha-1)^2 / \sum[k_j]$ | 0.39 | 0.25 |
| $(\beta-1)^2 / \sum[k_j]$ | 0.20 | 0.32 |
| $2\alpha r / \sum[n_j]$ | -0.44 | -0.27 |
| $-\alpha^2 / \sum[n_j]$ | 0.23 | 0.91 |
| $-\beta^2 / \sum[n_j]$ | 1.21 | 0.36 |

these watersheds is that Gulkana is drier and colder than Wolverine (Fig. 2 and A1). Although it is beyond the scope of the present work to establish precisely why the $\theta$ representation performs better for Gulkana and the CC representation performs better for Wolverine, it may be related to the way each module represents the timing of accumulation and melt.

The CC representation assumes that the snowpack is isothermal, which causes errors in how the internal energy is accounted

5  for and also impacts the net heat flux in the LST heat representation. The former point is related, in part, to the fact that the isothermal assumption neglects thermal conductivity within the snow. The thermal conductivity is lower for less dense snowpacks, which causes energy changes at the surface to propagate more slowly through the low density snowpack relative to a higher density equivalent snowpack. By this mechanism, the CC representation may overestimate internal energy deficit during dry and cold winter months, which would impact models applied to Gulkana more than models applied to Wolverine.

**Table 5.** Evaluation scores during model validation by mass flux module representation and watershed, averaged over heat module representation. The watershed listed is that which the model is validated for. Evaluation methodology and metrics are described in Secs. 2.5 and 3.3.

| Model | Gulkana | | | Wolverine | | |
|---|---|---|---|---|---|---|
| | Flow | SCA | Stake | Flow | SCA | Stake |
| $\theta$ | 0.66 | 0.79 | -0.41 | 0.31 | 0.54 | -0.32 |
| CC | 0.39 | 0.56 | 0.08 | 0.74 | 0.56 | 0.51 |

10  One characteristic of an improved mass flux representation would be that it allows snowpack temperature to vary vertically through the snowpack. This would have two advantages over the CC representation: (1) the net surface heat fluxes could be better calculated across climatic regimes and (2) the effect of the snowpack's thermal conductivity could be accounted for in

updating the snowpack's average internal energy. An approach to accomplish this would be to iteratively solve for the surface temperature by requiring that the surface energy balance be zero, as is done in Liston and Elder (2006a). Using such a method, melt would occur based on the energy balance at the surface of the snowpack rather than as a function of the snowpack's average internal energy, as is the case in the CC representation.

## 4.2 Regional Differences

If validation results are aggregated over model applications to the same region, it appears that on average the seven models better reproduce stake observations for Wolverine and better reproduce SCA observations for Gulkana. Simultaneously, though, they reproduce flow observations roughly equally well (Table 6). One important caveat is that averaging over models ignores intermodel differences in performance. To better understand causes of interregional model performance, we also calculate the linear correlation coefficient, $r$, between performance in each of the cryosphere observation types (i.e. SCA and stake) and flow, by region (Table 6). For Wolverine, the correlation between stake and flow performance is 0.99, while for Gulkana the correlation is not statistically significant. For Gulkana, the correlation between SCA and flow performance is 0.86, while for Wolverine the correlation is not statistically significant. One reason why stake performance may be more important to Wolverine is that a larger portion of the model domain and area contributing to the streamgage is glacier covered. It is surprising, though, that for Wolverine SCA performance is not significantly correlated to flow performance. We are not sure why SCA and flow performance are not correlated for Wolverine.

**Table 6.** Model-averaged evaluation performance scores and linear correlation coefficient, $r$, between performance of each cryosphere observation type and flow during model validation, shown by region. The watershed listed is that which the model is validated for. The top row for each watershed is the regionally averaged evaluation score; the first number in parentheses is $r$ and the second number is the p-value, which is an indicator of statistical significance (typically values less than 0.05 are considered significant). Evaluation methodology and metrics are described in Secs. 2.5 and 3.3.

| Region | Flow | SCA | Stake |
| --- | --- | --- | --- |
| Gulkana | 0.51 | 0.65 | -0.13 |
| | | (0.86; 0.012) | (0.43; 0.33) |
| Wolverine | 0.56 | 0.55 | 0.15 |
| | | (0.20; 0.66) | (0.99; 0.00) |

## 4.3 Model Accuracy, Precision, and Equifinality

The primary model assessment goal of this work is to evaluate the robustness of the seven conceptual cryosphere models between regions and climatic conditions. Two aspects of robustness are overall accuracy (i.e. predictive skill for each model application) and precision (i.e. how the predictive skill varies between applications of the model across regions and climatic conditions). Additionally, equifinality affects robustness in so far as different equally acceptable parameter sets selected through calibration may lead to different interpretations of model results. Tables B4 and B5 display results of the equifinality exercise

described in Sec. 3.3. In roughly half of the validation cases, the mean validation performance of the 100 best calibration parameter sets is slightly higher than the validation performance of the parameter set that performs best during calibration; the differences between these two metrics are almost always very small, though. Given that the assessment criteria includes flow, SCA, and stake observations and that the validation assessment is particularly stringent (i.e. applying the calibrated models to a different model domain with a different climatic regime), we believe this level of equifinality is acceptable. For example, the differences in model performance are smaller than those obtained in Finger et al. (2011) and Razavi and Tolson (2013), although because of differences in each assessment approach it is not possible to make quantitative comparisons.

Table 7 provides the validation performance of each model averaged across regions and the difference in performance between each region. Under this analysis the ETI(P)-CC and LST-CC models have the best predictive accuracy for reproducing flow and stake observations, while the SDI-$\theta$ and ETI(H)-$\theta$ models have the best predictive accuracy for reproducing SCA observations. The precision, i.e. how similar performance is between applications, for the ETI(P)-CC and LST-CC models is better than for the other five models, including for SCA performance. While there are minor differences in performance between the ETI(P)-CC and LST-CC models (e.g. with respect to stake performance), the differences in performance between these two models are relatively small compared to differences with the other five models. Therefore, we conclude that both the ETI(P)-CC and LST-CC models are superior to the other five models evaluated in terms of validation accuracy and precision.

**Table 7.** Regionally-averaged evaluation scores during model validation by model. Values in parentheses are absolute difference in model validation scores between regions. Evaluation methodology and metrics are described in Secs. 2.5 and 3.3.

| Model | Flow | SCA | Stake | Mean |
|---|---|---|---|---|
| SDI-$\theta$ | 0.58 | 0.68 | -0.04 | 0.41 |
| | (0.46) | (0.19) | (0.25) | (0.30) |
| SDI-CC | 0.28 | 0.47 | -0.20 | 0.18 |
| | (0.66) | (0.26) | (0.88) | (0.60) |
| ETI(H)-$\theta$ | 0.48 | 0.69 | -0.42 | 0.25 |
| | (0.26) | (0.21) | (0.29) | (0.25) |
| ETI(H)-CC | 0.50 | 0.54 | 0.31 | 0.45 |
| | (0.48) | (0.07) | (0.34) | (0.30) |
| ETI(P)-$\theta$ | 0.40 | 0.63 | -0.65 | 0.13 |
| | (0.32) | (0.34) | (0.23) | (0.30) |
| ETI(P)-CC | 0.75 | 0.60 | 0.51 | 0.62 |
| | (0.12) | (0.07) | (0.27) | (0.15) |
| LST-CC | 0.73 | 0.62 | 0.56 | 0.64 |
| | (0.12) | (0.12) | (0.20) | (0.15) |

The novel conceptual model formulations that we assess include the four that utilize the CC mass flux representation (i.e. the SDI-CC, ETI(H)-CC, ETI(P)-CC, and LST-CC models). If we compare only the pre-existing models (i.e. SDI-$\theta$, ETI(H)-

$\theta$, and ETI(P)-$\theta$), the SDI-$\theta$ model exhibits the best flow performance, the SDI-$\theta$ and ETI(H)-$\theta$ models exhibit the best SCA performance, and the SDI-$\theta$ model exhibits the best stake performance (Table 7). On average, the SDI-$\theta$ model outperforms the other models with $\theta$ mass flux representations, but none of these three pre-existing models is precise between watersheds. One significant difference between these three models is that the SDI-$\theta$ model relies only on temperature both for determining the heat flux and onset of melt conditions, while the ETI(H)-$\theta$ and ETI(P)-$\theta$ models use both temperature and shortwave radiation to determine the heat flux but only temperature to determine onset of melt conditions. The latter set of process representations is not physically consistent given that in reality onset of melt conditions is determined by the energy balance, which is impacted by all of the heat fluxes. This is not to claim that the SDI-$\theta$ model is more physically representative than the others, just that the heat and melt formulations may be more consistent.

## 4.4 Assessment Using Gulkana Only

In this section we provide results for a model assessment using only Gulkana watershed. The purpose of this is to understand the importance of the two watershed model assessment methodology (described in Sec. 3.3), in which the watersheds have contrasting climates and geographies, that we use to produce all of the model evaluation results provided elsewhere in this paper. For the Gulkana only assessment, we use the model parameter sets displayed in Table B1, which are calibrated for Gulkana from September 2000 through August 2010. We then validate these calibrated models for Gulkana from September 1990 through August 2000, using the preceding 36 months as a model a spin up period. Only stake and flow observations are available for this validation period. We cannot conduct a similar validation for Wolverine because stake observations are not available for this period.

The results of this exercise, displayed in Fig. 5, are intriguing because the relative ranking of model performance is somewhat different than that found through the two watershed assessment (e.g. Tab. 7). For the Gulkana only assessment, the ETI(H)-$\theta$ model exhibits the best validation flow and stake performance, followed by the ETI(P)-$\theta$ and SDI-$\theta$ models. This is consistent with the findings presented in Sec. 4.1 that the $\theta$ mass flux module representation performs better for Gulkana. In contrast to the main accuracy and precision results presented in Sec. 4.3, the ETI(P)-CC, and LST-CC models perform among the worst for the Gulkana only assessment. The contrast in results between the one watershed and two watershed assessments clearly highlights the importance of selecting an evaluation methodology that tests model behavior over the range of conditions that the model will be applied for. As an example, if the objective is to simulate recent historic conditions at Gulkana, the one watershed assessment is appropriate and finds that the ETI(H)-$\theta$ model should be used. If the model will be applied to altered climatic or geographic conditions, the two watershed assessment tests model performance over a wider range of conditions and finds that either the ETI(P)-CC or LST-CC models may be more suitable.

## 4.5 General Remarks and Future Directions

As discussed in Fujita and Ageta (2000), glacier surface processes are not sufficient to describe glaciers; for example, because up to 20% of melt refreezes. Surface refreezing is allowed in the CC mass flux module representation for snowpacks when the CC is positive, but not for glaciers. The primary differences between snow and ice heat fluxes in the models implemented

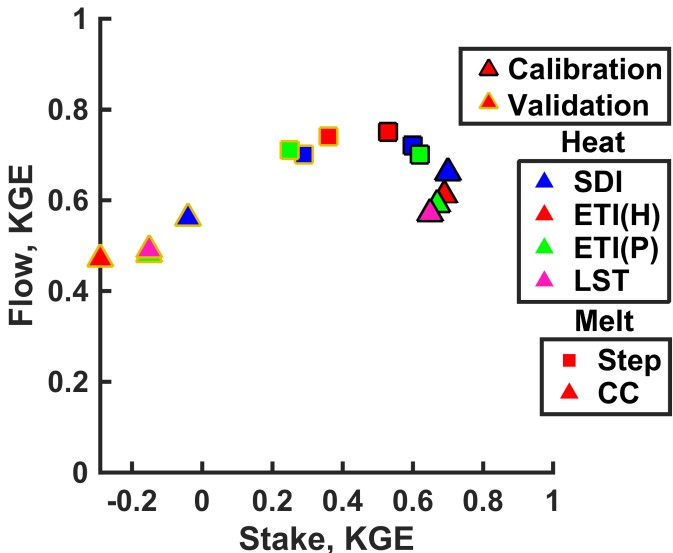

**Figure 5.** Model performance for Gulkana using Gulkana calibration (i.e. parameters from Table B1) and validating each model for Gulkana from September 1990 through August 2000. Only stake and flow observations are available during this validation period. The marker border indicates the whether the model period is calibration or validation, face color indicates heat module representation, and shape indicates mass flux module representation, as described in the legends.

here are that (1) ice has a different surface albedo than snow and (2) ice includes a fitting parameter to allow heat fluxes to translate into melt at a different rate than for snow (see Sec. 2.3). None of the models contains a representation of the ice's internal energy. Even SnowModel, an energy balance cryosphere hydrology model, does not characterize the internal energy of ice (Liston and Elder, 2006a). Yet, conceptual and energy balance cryosphere hydrology models are often applied to assess

the impacts of glacier melt on projected future streamflow. We therefore perceive this as a significant deficiency in current cryosphere hydrology models and an area where future work must be done.

None of the cryosphere hydrology models implemented here uses wind speed or humidity as inputs, but these variables are needed to explicitly represent the sensible and latent convective heat fluxes. The choice to exclude these inputs is made because they are less frequently observed for mountain environments, are difficult to characterize at high spatial resolutions,

and are therefore not commonly used in conceptual models. For continental snow and glacier environments, such as Gulkana, sublimation is an appreciable source of negative mass flux (Ohmura, 2001; DeWalle and Rango, 2008; Sicart et al., 2008). Similarly, sensible convection is important during the times of the year when the air and snow/ice surface temperatures differ from one another the most (e.g. late summer during the glacier ablation season). Each of the conceptual models we implement accounts for sensible heat fluxes through a degree-index term, which is a significant abstraction from the actual process. In

reality convective heat fluxes also depend on the wind speed and whether the atmosphere is in a stable, unstable, or neutral state (DeWalle and Rango, 2008). A potentially enlightening future experiment would be to develop a heat module representation that uses wind speed as an input and compare how uncertainties in characterizing wind speed impact representation of the heat

balance relative to using a degree-index term. The same experiment could be conducted for humidity to directly account for sublimation.

## 5  Conclusions

Understanding how the cryosphere will respond to climatic changes has important water resources implications and requires implementing models that are robust across geographic domains and climatic conditions. Conceptual cryosphere hydrology models are often used to make these types of projections in mountain environments due to data paucity, yet existing conceptual models are known to be less robust than physically-based model structures. We have developed the CCHF (Fig. 3) in order to systematically assess cryosphere modeling assumptions. We use the CCHF here to implement seven conceptual models (including existing and novel formulations; Table 3) for two glaciated watersheds in Alaska (Fig. 1). The CCHF enables us to interchange individual module representations, which provides more insight into the causes of differences in model performance compared to if we were to implement standalone models. While no single model outperforms the others for all categories of observations, the ETI(P)-CC and LST-CC models stand out as overall the most accurate and precise between climatic conditions and geographic domains when model performance is assessed using flow, SCA, and stake observations.

Our model analysis is by no means exhaustive, but provides some general insights and directions for future investigations. For example, we find that the $\theta$ mass flux module representation results in better model validation performance for Gulkana (the colder and drier of the two watersheds) and that the CC mass flux module representation results in better model validation performance under the same analysis for Wolverine (Table 5). Neither of these representations captures the impact of vertical temperature gradients within a snowpack on internal energy or on heat fluxes across the boundary. These deficiencies in cryosphere model formulations is consistent with other conceptual models (e.g. Hock, 1999, and Pellicciotti et al., 2005), and even with energy balance models such as SnowModel (Liston and Elder, 2006a) that are used in data-sparse mountain environments. Thus, we believe an area of future work with CCHF will be to better represent internal snow and glacier processes.

While the representations we assess are considered conceptual rather than physical, these model types exist on a spectrum rather than as discrete model types. We emphasize conceptual formulations in this work because this classification of processes representations is often used in data-sparse mountain regions. Of the representations we assess, we find that the more physically-based structures are more robust (i.e. overall the LST-CC and ETI(P)-CC models are the most robust). Therefore, interesting future work would be to incorporate more input data types, e.g. wind speed and vapor pressure, and assess the impacts of more physically-based heat flux representations on overall model accuracy and robustness. As we demonstrate through conducting assessments using one watershed and two watersheds with contrasting climatic conditions, no single model can be expected to perform the best under all circumstances. The reason is that each model's performance will depend on a host of factors, including climatic and geographic conditions, uncertainty in the required inputs, propagation of uncertainty through the model, and the evaluation methodology. Certain models, though, will inevitably perform better or worse under specific conditions, and the utility of the CCHF is that it enables users to easily test multiple cryosphere hydrology modeling hypothe-

ses for their system of interest. Our hope is that CCHF will be a useful tool for advancing our ability to represent cryosphere processes. We encourage interested parties to access CCHF version 1 on GitHub (distributed under 'thomasmosier/CCHF').

**Acknowledgments**

We would like to thank the Glumac Faculty Fellowship and the Oil Spill Recovery Institute for contributing funding to this project. We would also like to thank Anthony Arendt for his assistance in accessing the USGS stake measurements for Gulkana and Wolverine glaciers and Jordan Beamer for useful conversations about models and the Alaskan cryosphere.

**Appendix A: Joint Distribution of Precipitation and Temperature**

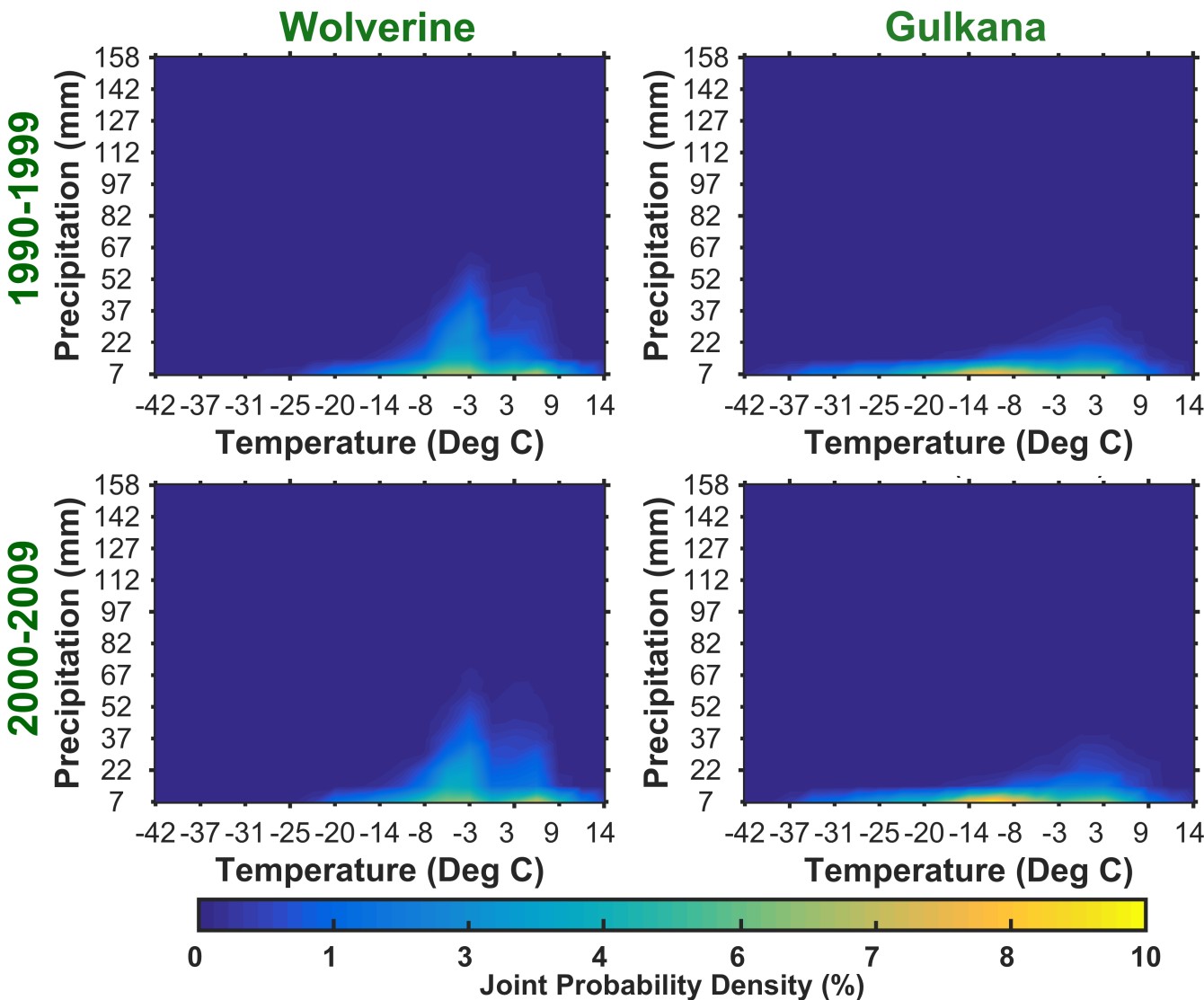

**Figure A1.** Joint distribution function of daily precipitation and mean surface air temperature for Gulkana and Wolverine watersheds (Fig. 1) from 1980-1989 and 1990-2009. The precipitation and temperature values are spatially averaged over each watershed. Climate representation is derived from the daily downscaled Climate System Forecast Reanalysis (Saha et al., 2010) product discussed in Sec. 3.1.

## B1  Calibrated Model Parameters

**Table B1.** All fitting parameters calibrated for Gulkana used in each of the models presented in Table B3.

| | SDI-$\theta$ | SDI-CC | ETI(H)-$\theta$ | ETI(H)-CC | ETI(P)-$\theta$ | ETI(P)-CC | LST-CC |
|---|---|---|---|---|---|---|---|
| $c_{atm1}$ (Table 2) | - | - | 0.98 | 0.82 | 0.69 | 0.74 | 0.61 |
| $c_{atm2}$ (Table 2) | - | - | 28.09 | 2.05 | 6.16 | 30.00 | 5.41 |
| $c_{fm}$ (Eqs. 8-11) | 8.20 | 14.57 | 28.09 | 27.98 | 2.27 | 17.95 | 13.87 |
| $c_{hoc,s}$ (Eq. 9) | - | - | 0.02 | 0.08 | - | - | - |
| $c_{hoc,i}$ (Eq. 9) | - | - | 4.69 | 2.28 | - | - | - |
| $c_{pel}$ (Eq. 10) | - | - | - | - | 0.27 | 0.42 | - |
| $c_{thr}$ (Eq. 13) | 3.92 | - | 3.80 | - | 3.62 | - | - |
| $c_{cc}$ (Eq. 14) | - | 0.10 | - | 0.20 | - | 0.20 | 0.27 |
| $c_{tsn}$ (Eq. 16) | - | - | - | - | - | - | 0.00 |
| $c_{slq}$ (Eq. 17) | 0.07 | 0.06 | 0.03 | 0.00 | 0.10 | 0.09 | 0.01 |
| $c_{glc}$ (Eq. 19) | 3.60 | 2.64 | 0.81 | 0.18 | 6.79 | 1.40 | 1.25 |
| $c_{dr}$ (Eq. 21) | 0.01 | 0.02 | 0.03 | 0.03 | 0.00 | 0.00 | 0.03 |
| $c_{smc}$ (Eq. 21) | 0.06 | 0.00 | 0.01 | 0.02 | 0.03 | 0.02 | 0.02 |
| $c_{pet}$ (Eq. 22) | 0.004 | 0.02 | 0.10 | 0.00 | 0.00 | 0.02 | 0.01 |
| $c_{tvl}$ (Eq. 23) | 0.02 | 0.29 | 0.06 | 0.20 | 0.08 | 0.38 | 0.29 |

**Table B2.** All fitting parameters calibrated for Wolverine used in each of the models presented in Table B3.

| | SDI-$\theta$ | SDI-CC | ETI(H)-$\theta$ | ETI(H)-CC | ETI(P)-$\theta$ | ETI(P)-CC | LST-CC |
|---|---|---|---|---|---|---|---|
| $c_{atm1}$ (Table 2) | - | - | 0.63 | 0.77 | 0.60 | 0.78 | 0.70 |
| $c_{atm2}$ (Table 2) | - | - | 7.23 | 9.42 | 30.00 | 14.10 | 30.00 |
| $c_{fm}$ (Eqs. 8-11) | 18.52 | 36.01 | 13.22 | 22.13 | 11.29 | 8.72 | 6.41 |
| $c_{hoc,s}$ (Eq. 9) | - | - | 0.05 | 0.14 | - | - | - |
| $c_{hoc,i}$ (Eq. 9) | - | - | 0.09 | 0.12 | - | - | - |
| $c_{pel}$ (Eq. 10) | - | - | - | - | 0.73 | 0.94 | - |
| $c_{thr}$ (Eq. 13) | 4.00 | - | 4.00 | - | 4.00 | - | - |
| $c_{cc}$ (Eq. 14) | - | 0.20 | - | 0.38 | - | 0.41 | 0.36 |
| $c_{tsn}$ (Eq. 16) | - | - | - | - | - | - | 0.08 |
| $c_{slq}$ (Eq. 17) | 0.08 | 0.02 | 0.10 | 0.04 | 0.07 | 0.07 | 0.07 |
| $c_{glc}$ (Eq. 19) | 1.15 | 0.51 | 0.94 | 0.45 | 1.56 | 1.49 | 1.11 |
| $c_{dr}$ (Eq. 21) | 0.00 | 0.00 | 0.03 | 0.00 | 0.02 | 0.00 | 0.00 |
| $c_{smc}$ (Eq. 21) | 0.08 | 0.37 | 0.13 | 0.02 | 0.05 | 0.64 | 0.36 |
| $c_{pet}$ (Eq. 22) | 0.03 | 0.00 | 0.09 | 0.02 | 0.02 | 3.32 | 0.01 |
| $c_{tvl}$ (Eq. 23) | 0.01 | 0.10 | 0.13 | 0.38 | 0.11 | 0.29 | 0.01 |

## B2  Calibration Performance Scores

**Table B3.** Best model evaluation scores during calibration. The top row for each model is one minus the PBE score for the 'SCA' columns and the KGE score for the 'Flow' and 'Stake' columns; the bottom row is the NSE score.

| Model | Gulkana | | | Wolverine | | |
|---|---|---|---|---|---|---|
| | Flow | SCA | Stake | Flow | SCA | Stake |
| SDI-$\theta$ | 0.72 | 0.63 | 0.60 | 0.74 | 0.64 | 0.65 |
| | (0.76) | (-) | (0.71) | (0.66) | (-) | (0.90) |
| SDI-CC | 0.66 | 0.58 | 0.70 | 0.70 | 0.51 | 0.77 |
| | (0.68) | (-) | (0.69) | (0.64) | (-) | (0.85) |
| ETI(H)-$\theta$ | 0.75 | 0.67 | 0.53 | 0.77 | 0.63 | 0.67 |
| | (0.74) | (-) | (0.71) | (0.58) | (-) | (0.89) |
| ETI(H)-CC | 0.61 | 0.61 | 0.69 | 0.73 | 0.52 | 0.80 |
| | (0.67) | (-) | (0.62) | (0.60) | (-) | (0.80) |
| ETI(P)-$\theta$ | 0.70 | 0.62 | 0.62 | 0.79 | 0.63 | 0.70 |
| | (0.76) | (-) | (0.66) | (0.63) | (-) | (0.91) |
| ETI(P)-CC | 0.59 | 0.59 | 0.67 | 0.80 | 0.59 | 0.73 |
| | (0.65) | (-) | (0.66) | (0.68) | (-) | (0.89) |
| LST-CC | 0.57 | 0.60 | 0.65 | 0.77 | 0.54 | 0.80 |
| | (0.64) | (-) | (0.67) | (0.68) | (-) | (0.88) |

## B3 Validation Performance Scores

**Table B4.** Best model evaluation scores during validation for Wolverine; parameter sets are calibrated for Gulkana. The top row for each model is one minus the PBE score for the 'SCA' columns and the KGE score for the 'Flow' and 'Stake' columns; the bottom row is the NSE score.

| Model | Flow | SCA | Stake |
|---|---|---|---|
| SDI-$\theta$ | $0.35; 0.38 \pm 0.02$ | $0.58; 0.58 \pm 0.01$ | $-0.16; -0.09 \pm 0.06$ |
| | $(0.56; 0.58 \pm 0.02)$ | (-) | $(0.42; 0.47 \pm 0.06)$ |
| SDI-CC | $0.61; 0.61 \pm 0.09$ | $0.60; 0.59 \pm 0.01$ | $0.24; 0.30 \pm 0.12$ |
| | $(0.69; 0.69 \pm 0.03)$ | (-) | $(0.68; 0.70 \pm 0.13)$ |
| ETI(H)-$\theta$ | $0.35; 0.34 \pm 0.00$ | $0.58; 0.57 \pm 0.00$ | $-0.27; -0.30 \pm 0.01$ |
| | $(0.55; 0.55 \pm 0.00)$ | (-) | $(0.23; 0.20 \pm 0.01)$ |
| ETI(H)-CC | $0.74; 0.73 \pm 0.01$ | $0.50; 0.50 \pm 0.00$ | $0.48; 0.38 \pm 0.04$ |
| | $(0.50; 0.49 \pm 0.01)$ | (-) | $(0.78; 0.72 \pm 0.02)$ |
| ETI(P)-$\theta$ | $0.24; 0.25 \pm 0.01$ | $0.46; 0.45 \pm 0.01$ | $-0.53; -0.51 \pm 0.02$ |
| | $(0.47; 0.47 \pm 0.01)$ | (-) | $(-0.07; -0.07 \pm 0.02)$ |
| ETI(P)-CC | $0.81; 0.80 \pm 0.03$ | $0.56; 0.56 \pm 0.01$ | $0.64; 0.63 \pm 0.11$ |
| | $(0.68; 0.69 \pm 0.03)$ | (-) | $(0.79; 0.81 \pm 0.10)$ |
| LST-CC | $0.79; 0.78 \pm 0.02$ | $0.56; 0.56 \pm 0.00$ | $0.66; 0.67 \pm 0.04$ |
| | $(0.67; 0.66 \pm 0.01)$ | (-) | $(0.79; 0.80 \pm 0.04)$ |

**Table B5.** Best model evaluation scores during validation for Gulkana; parameter sets are calibrated for Wolverine. The top row for each model is one minus the PBE score for the 'SCA' columns and the KGE score for the 'Flow' and 'Stake' columns; the bottom row is the NSE score.

| Model | Flow | SCA | Stake |
|---|---|---|---|
| SDI-$\theta$ | $0.81; 0.78 \pm 0.03$ | $0.77; 0.78 \pm 0.01$ | $0.09; -0.07 \pm 0.10$ |
| | $(0.63; 0.63 \pm 0.02)$ | (-) | $(0.52; 0.34 \pm 0.12)$ |
| SDI-CC | $-0.05; -0.15 \pm 1.00$ | $0.34; 0.36 \pm 0.06$ | $-0.64; -0.88 \pm 2.53$ |
| | $(0.08; -0.92 \pm 10.00)$ | (-) | $(-0.94; -6.11 \pm 50.81)$ |
| ETI(H)-$\theta$ | $0.61; 0.63 \pm 0.05$ | $0.79; 0.79 \pm 0.00$ | $-0.56; -0.55 \pm 0.15$ |
| | $(0.50; 0.52 \pm 0.05)$ | (-) | $(-0.28; -0.27 \pm 0.20)$ |
| ETI(H)-CC | $0.26; 0.29 \pm 0.04$ | $0.57; 0.58 \pm 0.01$ | $0.14; 0.17 \pm 0.05$ |
| | $(0.49; 0.52 \pm 0.04)$ | (-) | $(0.22; 0.25 \pm 0.05)$ |
| ETI(P)-$\theta$ | $0.56; 0.59 \pm 0.05$ | $0.80; 0.80 \pm 0.00$ | $-0.76; -0.72 \pm 0.14$ |
| | $(0.47; 0.51 \pm 0.05)$ | (-) | $(-0.54; -0.48 \pm 0.20)$ |
| ETI(P)-CC | $0.69; 0.69 \pm 0.15$ | $0.63; 0.64 \pm 0.04$ | $0.37; 0.20 \pm 0.30$ |
| | $(0.69; 0.69 \pm 0.08)$ | (-) | $(0.66; 0.28 \pm 0.28)$ |
| LST-CC | $0.67; 0.63 \pm 0.21$ | $0.68; 0.68 \pm 0.06$ | $0.46; 0.05 \pm 0.40$ |
| | $(0.74; 0.67 \pm 0.14)$ | (-) | $(0.65; 0.20 \pm .40)$ |

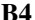

**Figure B1.** Model flow performance during validation as a function of SCA and stake performance. The marker border indicates phase of assessment, face color indicates heat module representation, and shape indicates mass flux module representation, as described in the legends. Subplot A displays KGE scores for stake measurements and flow. Subplot B displays one minus the PBE score for SCA and the KGE score for flow.

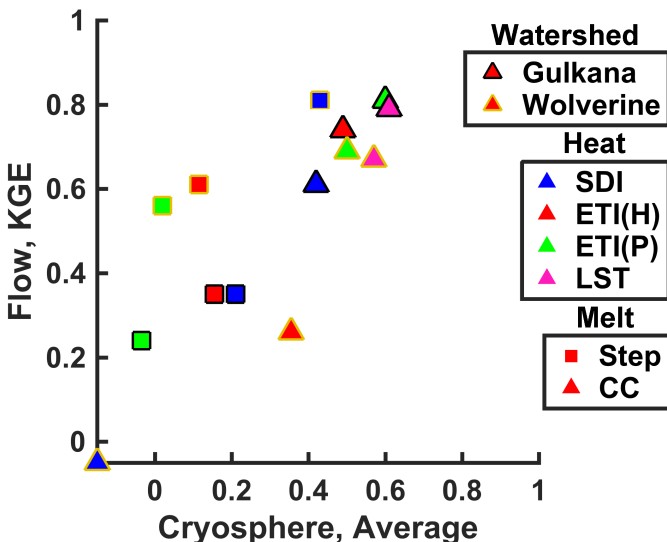

**Figure B2.** Model flow performance during validation as a function of combined cryosphere performance using the methodology and metrics described in Secs. 2.5 and 3.3. The marker border indicates the validation watershed, face color indicates heat module representation, and shape indicates mass flux module representation, as described in the legends.

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
