# Peer review of "How Much Cryosphere Model Complexity is Just Right? Exploration Using the Conceptual Cryosphere Hydrology Framework"

_The Cryosphere, 2016_

## Referee Comment (RC1) · Anonymous Referee #1 · 3 Apr 2016

This MS presents a model framework for simulating snow and glacier melt and related runoff formation. A number of established and partly novel algorithms are implemented and in a calibration-validation study for two different catchments; the authors assess the value of individual concepts. Available models of snow and ice melt almost form a continuum in terms of complexity and data demands and a study shedding light on how much complexity actually is needed, and would come as a highly welcome guidance for the modelling community.

Initially, I was intrigued by the title which poses this timely and relevant question. However, while reading, I became disappointed and find the MS hampered by several, severe deficiencies. These are related to the methodology, the way how the background

is presented, and the discussion of the question asked in the title. In total, the results do not provide sufficient support for the conclusions.

Major comments:

The calibration of conceptual models is often plagued by equifinality, i.e. that several different parameter combinations yield undistinguishable performance. This is a typical problem for models that experience error compensation, such as here. Several 'best performers' will reveal different behavior in a forecast/ validation situation. Hence, picking a single parameter combination, out of potentially several equally valid ones, may yield a low performance when validating, while other members of the ensemble of 'best performers' may perform well. It is misleading to neglect this parameter uncertainty and instead to make when the model concept responsible for the low performance. This shortcoming does compromise the entire conclusions about different concepts. Here, the authors either have to demonstrate that the performance topography has a sharp peak and thereby support the usage of a singular best performing parameter combination. Alternatively, an ensemble of similar best performers should be evaluated to explore the range of uncertainty when validating.

The way how SCA is evaluated is not very useful: the 3 dimensional information (2 spatial dims, 1 temporal) is reduced to a single one. the daily MODIS product lacks spatial completeness which is addressed by temporally aggregating the SCA maps for each month. It is left unclear which timestamp is associated to this aggregated SCAmap and how this is compared to the model results. This way much of the temporal information in the data is lost. In addition, the spatial information is removed by spatially averaging SCA over the domain. This leaves us with a performance measure that is most likely almost insensitive to whatever the model is doing, provided that the model reproduces the snow seasonality. However, severe model misbehavior such as snow in the wrong part of the catchment or missing meltout by 10 days is not at all penalized by averaging procedure.
Confusing presentation of different models implemented, what is new, what not? The original Hock 1999 algorithm does not include albedo but employs 2 different radiation parameters to account for differences between snow and ice. The stated equation here (eq2) is hence not identical to the original but a variant of it. This needs to be clearly stated.

Misleading statement about novelty of the CC approach. Your argumentation is circular by stating P4 L25: "a common concept in snow process modelling..." and then later claim "we are not aware of any previous conceptual cryosphere models that incorporate internal energy or 'negative melt' into their formulation". Isn't that a contradiction?

Detailed comments:

It is inappropriate to re-define units of a variable (P9 L3), either introduce a different variable (which would be confusing) or state how it is converted from one unit to another.

Units confusion: for instance M should carry units [m s-1] in water equivalents and is already a specific quantity and does not require repeated normalisation! So the units used here [m/m2 = m-1] are wrong. This applies several places throughout the MS.

Confusing notation: delta is used as operator (e.g. in delta t) but also as concept name (?)

How is the CC determined? Eq 4 suggests it is a function of T but then eq 11 makes Ts a function of CC, isn't that circular?

Redefining variables, eq 17 defines delta t as travel time whereas previously it was used as time step (P9 L13)

Eq 13: I do not get the meaning of H_ice/H, why is that needed? The difference between snow and ice melt would be accounted for by the difference in albedo when calculating H, so what effect is accounted for by this factor?

[Figure]

Eq 15: the units do not work out here, runoff should be in [m3 s-1], but then d_r needs
to be converted from [days-1] to [s-1]

---

## Referee Comment (RC2) · Anonymous Referee #2 · 4 May 2016

**General comments**

The authors describe the development and application of a modular framework combining several conceptual cryosphere process modules. In the paper, they focus on the modules for simulating heat transfer and mass and internal energy of snow and ice. The model is calibrated for two glacierized catchments in Alaska and validated (in the respective other catchments) for seven model combinations using MODIS snow cover products, stake measurements, and runoff observations. Results show that the more physically based methods tend to be more reliable, however no single best module combination can be identified.

I like the concept of a modular open source framework for simulating cryosphere pro-

cesses in data-sparse environments very much and also like the general structure of the paper. However, there are some issues in the manuscript which should be addressed by the authors.

Probably the biggest issue I see, which possibly also affects the conclusions, is the way the MODIS validation procedure is performed. First, both the spatial and the temporal resolution of the chosen MODIS product is, in my opinion, unnecessarily coarse. 0.05deg are approx. 5 km, hence I assume only very few MOD10CM pixels are within the catchment boundaries? On top of that, by additionally averaging the monthly MODIS SCA pixels over the catchment area, even more of the original information is lost (as no pixel-by-pixel comparison is performed). A more valuable validation strategy would have been to use the daily (or possibly 8-daily) 500 m MODIS products (while applying a cloud cover threshold), and to use another skill score (e.g. the fraction of correctly classified pixels) besides just comparing watershed-averaged SCA values. Otherwise, the distributed nature of both the MODIS data and the model results is neglected by lumping the results together into a single number. Additionally, in my experience, the MODIS snow mapping algorithm generally also classifies ice surfaces as "snow-covered". Since large parts of both of the investigated catchments are glacierized, I would assume that in these pixels MODIS and the model results always match, leading to a positive bias in the SCA validation results?

Additionally, the authors should check the units of the variables more carefully. For example, throughout the manuscript several times units of "$m^{-1}$" are used (e.g., for M, CC, SWE, snlr, iclr, SM, SMC), which should in fact be m (or m $s^{-1}$). Also, in several equations the units do not work out.

**Specific comments**

- Is there a reason for the comparatively coarse spatial resolution (30 arcseconds, i.e. approx. 1 km?), considering that the investigated catchments are relatively small? This might be at least one reason for the generally quite poor skill scores

for the stake measurement comparison (especially with regard to the bias, as shown in Table 3), due to the considerable scale differences between a single point on a glacier and a 1 km$^2$ model pixel.

- I like the approach of evaluating the robustness of the model by calibrating it for one watershed and validating it in the other, however it would have probably been very insightful if the calibrated model would have also been applied and evaluated in the same catchment (using a split-sample test) prior to transfering the parameters to the other one. I assume this has not been done due to the lack of sufficiently long validation data time series?

- The authors state that the CCHF is open source and available to interested parties, which I very much appreciate. However, there is no mention on how/where to obtain the source code. I would suggest to add this information to the manuscript.

- Section 1.2: Besides the description of the climates, possibly add some more general information about the two catchments (e.g. area, elevation range, glacierization, ...).

- Section 2: In the introduction of the section, I would suggest to add a sentence about the temporal (i.e. daily) and spatial resolutions the model is/can be applied on, as this information appears only later in section 3.1. Besides, some remarks about the meteorological variables that are used in the model (especially which variables are required as input data (minimum/maximum/mean temperature and precipitation?) and which are calculated/parameterized (shortwave and longwave radiation?)) could be added, as this is not immediately clear from the manuscript. Additionally, what is missing from the model description is information about the precipitation-phase partitioning method(s) available in the model, and if any kinds of precipitation adjustment functions (e.g. for gauge undercatch) are implemented?

- Eq. (13): It is not immediately clear to me why the scaling of potential ice melt with $c_g H_{ice}/H$ is necessary. I would assume that the differences in energy required to melt snow vs. ice are already accounted for by the differences in albedo, which is taken into account in all heat transfer formulations except SDI (and in this case this could be overcome by introducing two separate degree-index factors for snow and ice, respectively)?

- Eqs. (14-15): Please introduce $r_f$ after eq. (14) (where it first appears) instead of after eq. (15). The units of $r_f$ do not work out in eqs. (14) and (15) (in eq. (14) it would have to be in m, while according to eq. (15) it is in m$^3$). Additionally, in eq. (14) $r_{f,i}$ should likely be $r_{f,i-1}$, otherwise there would be a circular reference?

- P17L3: Stating that glacier models are commonly evaluated only for a few days (!) is probably an exaggeration – I think it is well established that multi-year evaluation periods are necessary for glaciological purposes.

**Technical corrections**

- There is a typo in the title ("Cryrosphere")

- P4L10: $[f_s]$ should be $[f_m]$, in the units again m instead of m$^{-1}$

- P4L24: ti → to

- P5L1: latent heat → latent heat of fusion

- P5L3: i.e → i.e.

- P5L29: (Simpson et al., 2002) → Simpson et al. (2002)

- Table 1: $\alpha_o$ (from the table heading) never actually appears in the table

- P9L3: m$^{-}2 \rightarrow$ m$^{-2}$

- P9L24: I would move this sentence ("... where the negative bounds on CC is zero") a few lines up (after eq. (9)).

- P10L17: "the the"

- P11L20: $c_t$ should probably be $t_l$?

- P15L6: their $\rightarrow$ there

- P18L21-22: "has the predictive skill" $\rightarrow$ "has the best predictive skill"? (2x)

- Table 8 (heading): vise $\rightarrow$ vice

---

## Author Comment (AC1) · 1 Aug 2016

We thank the anonymous reviewers for their thorough reviews. We have made many significant changes to the manuscript in response and believe the analysis is much stronger for it.

We would also like to take this opportunity to reiterate a point made in our manuscript, which is that which model is "superior" depends on a host of factors (e.g. errors in input data, spatial and temporal resolution, evaluation criteria). We therefore do not purport to have identified a model structure that will perform the best under any circumstance. Instead, we believe the principal benefit of this work is the development of a tool for systematically assessing various cryosphere model formulations for the system(s) and

evaluation criteria of interest. Using this framework, we believe that some informative conclusions can be drawn for the models and test cases we evaluate.

Noting this, we agree that strong standards must be upheld in evaluating models, and have strengthened our analysis in three main ways: (1) Evaluating the impact of parameter uncertainty (i.e. equifinality) on results and (2) Comparing the spatial pattern of snow covered area captured by MODIS to model simulations of snow. Additionally, we now use 8-day MODIS images instead of the monthly values used in the original manuscript. (3) Conducting a one watershed validation exercise to complement our primary analysis (i.e. the two watershed approach), in which the same models are calibrated and validated using only Gulkana. In order to address the concerns of the two reviewers, we have undertaken significant additional model development and have rerun all of the model simulations used in the manuscript.

Attached, please find the our point-by-point response to each of the reviewers' comments, including explanations of how we addressed the points in our revised manuscript.

Kind regards, Thomas Mosier (on behalf of David Hill and Kendra Sharp)

Please also note the supplement to this comment:
http://www.the-cryosphere-discuss.net/tc-2016-17/tc-2016-17-AC1-supplement.pdf

**Supplement:**

Response to Reviewers, TCD Manuscript

"How Much Cryosphere Model Complexity is Just Right? Exploration Using the Conceptual Cryosphere Hydrology Framework"

Mosier et al.

1 August 2016

We thank the anonymous reviewers for their thorough reviews. We have made many significant changes to the manuscript in response and believe the analysis is much stronger for it.

We would also like to take this opportunity to reiterate a point made in our manuscript, which is that which model is "superior" depends on a host of factors (e.g. errors in input data, spatial and temporal resolution, evaluation criteria). We therefore do not purport to have identified a model structure that will perform the best under any circumstance. Instead, we believe the principal benefit of this work is the development of a tool for systematically assessing various cryosphere model formulations for the system(s) and evaluation criteria of interest. Using this framework, we believe that some informative conclusions can be drawn for the models and test cases we evaluate.

Noting this, we agree that strong standards must be upheld in evaluating models, and have strengthened our analysis in three main ways:

(1) Evaluating the impact of parameter uncertainty (i.e. equifinality) on results and
(2) Comparing the spatial pattern of snow covered area captured by MODIS to model simulations of snow. Additionally, we now use 8-day MODIS images instead of the monthly values used in the original manuscript.
(3) Conducting a one watershed validation exercise to complement our primary analysis (i.e. the two watershed approach), in which the same models are calibrated and validated using only Gulkana.

In order to address the concerns of the two reviewers, we have undertaken significant additional model development and have rerun all of the model simulations used in the manuscript.

Below are our responses to all review comments. The original reviewer comments are in plain text and author responses are in **bold**. Text added to the manuscript is provided in ***bold italic*** font. References to specific lines have the format PxLy, where the x refers to the page number and y to the line. Thank you for your consideration of our revised manuscript.

Kind Regards,

Thomas M. Mosier, David F. Hill, and Kendra V. Sharp

**REVIEWER 1:**

This MS presents a model framework for simulating snow and glacier melt and related runoff formation. A number of established and partly novel algorithms are implemented and in a calibration-validation study for two different catchments; the authors assess the value of individual concepts. Available models of snow and ice melt almost form a continuum in terms of complexity and data demands and a study shedding light on how much complexity actually is needed, and would come as a highly welcome guidance for the modelling community.

Initially, I was intrigued by the title which poses this timely and relevant question. However, while reading, I became disappointed and find the MS hampered by several, severe deficiencies. These are related to the methodology, the way how the background is presented, and the discussion of the question asked in the title. In total, the results do not provide sufficient support for the conclusions.

**As we note below in response to your specific comments, we have addressed the methodological issues that you raise. We have also rewritten the introduction (Section 1) and believe that it clearly outlines the background, concepts relevant to the work, and objectives. In particular, we note that developing a modeling framework for addressing these types of questions is on its own a significant undertaking. We do not purport to arrive at a conclusive answer regarding optimal cryosphere model complexity, and fundamentally believe that there is not objectively correct answer that holds across all cryosphere systems. Instead, as our title suggests, we explore the question of conceptual cryosphere complexity. As stated in our updated introduction (see P4L5-8),** "*The specific objectives of this study are to (1) demonstrate that CCHF is a useful tool for developing novel conceptual cryosphere hydrology models and (2) explore differences in accuracy and precision of existing and novel conceptual cryosphere hydrology models for two well-monitored glaciated model domains.*" **We believe that our updated manuscript achieves these objectives.**

**Major comments:**

The calibration of conceptual models is often plagued by equifinality, i.e. that several different parameter combinations yield undistinguishable performance. This is a typical problem for models that experience error compensation, such as here. Several 'best performers' will reveal different behavior in a forecast/ validation situation. Hence, picking a single parameter combination, out of potentially several equally valid ones, may yield a low performance when validating, while other members of the ensemble of 'best performers' may perform well. It is misleading to neglect this parameter uncertainty and instead to make when the model concept responsible for the low performance. This shortcoming does compromise the entire conclusions about different concepts. Here, the authors either have to demonstrate that the performance topography has a sharp peak and thereby support the usage of a singular best performing parameter combination. Alternatively, an ensemble of similar best performers should be evaluated to explore the range of uncertainty when validating.

**We agree with the above point and have updated the results to include aggregated performance statistics of the 100 sets of fitting parameters that perform best during model calibration. As stated in Section 3.3 (P20L28-P21L2),** "*We assess equifinality due to the cryosphere process representations by conducting validation on the 100 parameter sets that perform best during the first stage of calibration. We set the fitting parameters corresponding to non-cyrosphere parameters (i.e.\ those calibrated in the second stage) to their calibrated values. Thus, we are only investigating equifinality in fitting parameters related to modeling cryosphere processes. We provide equifinality assessment statistics in all tables of validation run results. Specifically, we include (1) validation performance for the parameter set that performs best during calibration, (2) the mean validation performance for the 100 best performing calibration runs, and (3) the standard deviation of validation performance for the 100 best performing calibration runs. The general format we use for reporting these statistics is `$x_{1}$; $x_{2} \pm x_{3}$', where the $x_{i}$ refer to the three numbers statistics from the previous sentence.*" **Equifinality statistics are contained in Tables B4 and B5. These results are discussed on P24L21-P25L7.**

The way how SCA is evaluated is not very useful: the 3 dimensional information (2 spatial dims, 1 temporal) is reduced to a single one. the daily MODIS product lacks spatial completeness which is addressed by temporally aggregating the SCA maps for each month. It is left unclear which timestamp is associated to this aggregated SCA map and how this is compared to the model results. This way much of the temporal information in the data is lost. In addition, the spatial information is removed by spatially averaging SCA over the domain. This leaves us with a performance measure that is most likely almost insensitive to whatever the model is doing, provided that the model reproduces the snow seasonality. However, severe model misbehavior such as snow in the wrong part of the catchment or missing meltout by 10 days is not at all penalized by averaging procedure.

**We have addressed this by using MOD10A2, which has an 8-day temporal resolution and 500 m spatial resolution. As explained in Section 2.5 (P16L31-P17L28), we assess errors in the distribution of modeled snow water equivalent relative to the MODIS images according to the method developed by Barakya and Poschl (2008). This method captures errors in both the spatial and temporal patterns, where the spatial errors are calculated at the model resolution and the temporal errors are calculated at the 8-day resolution of the MODIS images.**

Confusing presentation of different models implemented, what is new, what not? The original Hock 1999 algorithm does not include albedo but employs 2 different radiation parameters to account for differences between snow and ice. The stated equation here (eq2) is hence not identical to the original but a variant of it. This needs to be clearly stated.

**We have restructured Sections 1.1, 1.2, 2.1, and 2.2 to more clearly present background on relevant model structures and the model formulations implemented in our work. We have also chosen to update the Hock model presentation (Section**

**1.1; Eq. 2) and heat formulation based on Hock's model (Eq. 9) to be consistent with the model presented in Hock (1999).**

Misleading statement about novelty of the CC approach. Your argumentation is circular by stating P4 L25: "a common concept in snow process modelling. . ." and then later claim "we are not aware of any previous conceptual cryosphere models that incorporate internal energy or 'negative melt' into their formulation". Isn't that a contradiction?

**Our language was imprecise in this instance. Our intent was to convey that cold content (CC) is a common concept (in general terms) but that to our knowledge the concept has not been applied in a distributed conceptual cryosphere computational models. The updated statement in Section 1.2 (P7L3-L4) is "*we are not aware of any previous distributed cryosphere computational models that incorporate internal energy or `negative melt' into their formulation using CC.*"**

Detailed comments:

It is inappropriate to re-define units of a variable (P9 L3), either introduce a different variable (which would be confusing) or state how it is converted from one unit to an- other.

**In updating our model presentation (Sections 1.1, 1.2, 2.1, and 2.2), we have edited the definitions of each variable such that the units for a given variable are consistent throughout the manuscript.**

Units confusion: for instance M should carry units [m s-1] in water equivalents and is already a specific quantity and does not require repeated normalisation! So the units used here [m/m2 = m-1] are wrong. This applies several places throughout the MS.

**We updated the units of variables that previously included a factor of "$m^{-1}$" such that they are now "m".**

Confusing notation: delta is used as operator (e.g. in delta t) but also as concept name (?)

**We have left the capital delta to denote a step in time (e.g. Eq. 1) and replaced the mass flux concept name (e.g. Eq. 13) with the lowercase theta, which is a symbol that is sometimes used to denote the Heaviside Step Function.**

How is the CC determined? Eq 4 suggests it is a function of T but then eq 11 makes Ts a function of CC, isn't that circular?

**No, this is not circular because Eq. 4 can straightforwardly be inverted to instead solve for snow temperature (assuming an isothermal snowpack). We note, however, that our presentation of the CC as a mass flux module may be confusing given that we**

**update the CC based on melt potential. We therefore have changed the symbol for the cold content in the CC mass flux representation (Eqs. 14-16), replacing "CC" with "$w_{c}$".**

Redefining variables, eq 17 defines delta t as travel time whereas previously it was used as time step (P9 L13)

**Thank you. We have changed the symbol for travel time in Eq. 17 and elsewhere to $t_{t}$.**

Eq 13: I do not get the meaning of H_ice/H, why is that needed? The difference between snow and ice melt would be accounted for by the difference in albedo when calculating H, so what effect is accounted for by this factor?

The reason for this factor is that net heat at the snow surface and net heat at the ice surface are calculated for each grid cell with snow or ice, respectively, during each time step. In the case where a cell has both snow and ice, snow is allowed to melt first and any remaining energy goes into melting ice (during the same time step). Therefore, $H_{ice}/H$ is used to scale the melt energy. We have updated the text in Section 2.3 to clarify this point: "The purpose of Eq. 19 is that it allows ice melt to occur in the same time step in which the snowpack has completely melted from a grid cell. Eq. 19, which is essentially a ratio to scale the melt potential available for ice, approximates the portion of energy that is still available for ice melt after all snow has melted during the current time step. Over short time steps, the error incurred by not allowing snow and ice melt to occur for a given grid cell during the same time step is small; however, including Eq. 19 allows the model to scale better between small and large time steps."

Eq 15: the units do not work out here, runoff should be in [m3 s-1], but then d_r needs to be converted from [days-1] to [s-1]

**We have corrected the units related to Eq. 21 in the revised manuscript.**

**REVIEWER 2:**

**General comments**

The authors describe the development and application of a modular framework combining several conceptual cryosphere process modules. In the paper, they focus on the modules for simulating heat transfer and mass and internal energy of snow and ice. The model is calibrated for two glacierized catchments in Alaska and validated (in the respective other catchments) for seven model combinations using MODIS snow cover products, stake measurements, and runoff observations. Results show that the more physically based methods tend to be more reliable, however no single best module combination can be

identified.

I like the concept of a modular open source framework for simulating cryosphere processes in data-sparse environments very much and also like the general structure of the paper. However, there are some issues in the manuscript which should be addressed by the authors.

Probably the biggest issue I see, which possibly also affects the conclusions, is the way the MODIS validation procedure is performed. First, both the spatial and the temporal resolution of the chosen MODIS product is, in my opinion, unnecessarily coarse. 0.05deg are approx. 5 km, hence I assume only very few MOD10CM pixels are within the catchment boundaries? On top of that, by additionally averaging the monthly MODIS SCA pixels over the catchment area, even more of the original information is lost (as no pixel-by-pixel comparison is performed). A more valuable validation strategy would have been to use the daily (or possibly 8-daily) 500 m MODIS products (while applying a cloud cover threshold), and to use another skill score (e.g. the fraction of correctly classified pixels) besides just comparing watershed-averaged SCA values. Otherwise, the distributed nature of both the MODIS data and the model results is neglected by lumping the results together into a single number. Additionally, in my experience, the MODIS snow mapping algorithm generally also classifies ice surfaces as "snow-covered". Since large parts of both of the investigated catchments are glacierized, I would assume that in these pixels MODIS and the model results always match, leading to a positive bias in the SCA validation results?

**As noted in response to a similar comment from Reviewer 1, we have completely overhauled the snow covered area assessment. Changes include that we now use the (1) MOD10A2 MODIS dataset, which has a 500 m spatial resolution and 8-day temporal resolution and (2) assessment strategy by Barakya and Poschl (2008).**

**We agree with your comment that MODIS often classifies ice as snow. Therefore, we remove all pixels that are classifies as snow covered or lake ice in more than 90% of all MODIS images considered from analysis (see P17L12-L13). Given that a large percentage of the model domains are glacier covered, this criterion removes half of the model grid cells from consideration in both watersheds (see Fig. 1 and Table 1).**

Additionally, the authors should check the units of the variables more carefully. For example, throughout the manuscript several times units of "$m^{-1}$" are used (e.g., for M, CC, SWE, snlr, iclr, SM, SMC), which should in fact be m (or $m\ s^{-1}$). Also, in several equations the units do not work out.

**We have converted units with factors of $m^{-1}$ to $m$ throughout. We have also edited the units in several equations to ensure units for each variable are consistent throughout the manuscript and that units are balanced.**

**Specific comments:**

Is there a reason for the comparatively coarse spatial resolution (30 arcseconds, i.e. approx. 1 km?), considering that the investigated catchments are relatively small? This might be at least one reason for the generally quite poor skill scores for the stake measurement comparison (especially with regard to the bias, as shown in Table 3), due to the considerable scale differences between a single point on a glacier and a 1 km$^2$ model pixel.

**Given the small size of the catchments, it would be possible to evaluate the model at a finer spatial resolution; however, this would increase computational time by the square of the increase in resolution. The reason we use 30 arcseconds is that the calibration routine involves thousands of model runs and multiple models are being tested. Increasing the spatial resolution would likely reduce errors associated with calculating shortwave radiation, since it depends on slope and aspect. However, based on other modeling studies that include Wolverine and Gulkana glaciers (e.g. Beamer et al. 2016), as well as our own experience in the present study, we believe significant sources of uncertainty are the climate inputs (here we use CFSR by NCEP). Evaluating the model at a finer resolution would require resampling or downscaling the climate data to the same finer resolution, but would not likely reduce these climate uncertainties.**

**We note also that 30 arcseconds is a common spatial resolution for implementing distributed models at.**

I like the approach of evaluating the robustness of the model by calibrating it for one watershed and validating it in the other, however it would have probably been very insightful if the calibrated model would have also been applied and evaluated in the same catchment (using a split-sample test) prior to transfering the parameters to the other one. I assume this has not been done due to the lack of sufficiently long validation data time series?

**Two reasons we use the 10 year, two watershed validation strategy as our primary assessment are that (1) we want to reduce equifinality, which necessitates using a calibration period that captures a wide variety of conditions and multi-objective evaluation criteria and (2) the USGS observations have significant gaps at both glaciers, but especially at Wolverine, for the years prior to 2000. Points 1 and 2 necessitate using the years 2000-2010 because these are the years that we have all three observation variables and CFSR climate data.**

**Additionally, one of our primary objectives of the study is to investigate how robust each model is across climates and geographies. Our assessment strategy of calibrating for one watershed and validating for the other is the best means of achieving this because there is more variation in climate between the two watersheds than for the same watershed over two successive decades (see Fig. 5 of the climates for 1990-1999 and 2000-2009).**

**We have, however, conducted an additional assessment using only Gulkana**

**watershed (Section 4.4). The calibration is the same as in our two watershed assessment (i.e. we simply use the calibrated parameter sets from the Gulkana calibration), but we then validate the models for September 1990 through August 2000. This exercise indeed, turns out to yield interesting results because the relative ranking of models differs for the Gulkana only assessment compared to the two watershed assessment. The main conclusion of this additional assessment is that it is important to design the evaluation methodology to best represent the conditions that the model will be applied for. For example, the results of the Gulkana only assessment would be potentially misleading if the objective were to simulate conditions under future climate change or over a large ungagged basin. If, though, the objective were to simulate recent historic conditions at Gulkana, the two watershed assessment would also not result in the optimal model choice.**

The authors state that the CCHF is open source and available to interested parties, which I very much appreciate. However, there is no mention on how/where to obtain the source code. I would suggest to add this information to the manuscript.

**If this manuscript is accepted to the Cryosphere, we will release the CCHF code to GitHub and will update the manuscript accordingly.**

Section 1.2: Besides the description of the climates, possibly add some more general information about the two catchments (e.g. area, elevation range, glacierization, …).

**We have added the requested information on the two model domains in the form of Table 1  (Section 1.3 in revised manuscript).**

Section 2: In the introduction of the section, I would suggest to add a sentence about the temporal (i.e. daily) and spatial resolutions the model is/can be applied on, as this information appears only later in section 3.1. Besides, some remarks about the meteorological variables that are used in the model (especially which variables are required as input data (minimum/maximum/mean temperature and precipitation?) and which are calculated/parameterized (shortwave and longwave radiation?)) could be added, as this is not immediately clear from the manuscript. Additionally, what is missing from the model description is information about the precipitation-phase partitioning method(s) available in the model, and if any kinds of precipitation adjustment functions (e.g. for gauge undercatch) are implemented?

**We have updated Section 2 (P8L30-P11L6) to include the requested information, including applicable spatial and temporal resolutions, climate inputs, and precipitation partitioning. We have also added text to Section 2 (P8L28-L29) explaining what type of information is included in Sections 2 and 3, respectively.**

Eq. (13): It is not immediately clear to me why the scaling of potential ice melt with

$c_g H_{ice}/H$ is necessary. I would assume that the differences in energy required to melt snow vs. ice are already accounted for by the differences in albedo, which is taken into account in all heat transfer formulations except SDI (and in this case this could be overcome by introducing two separate degree-index factors for snow and ice, respectively)?

**See response on similar question by Reviewer 1. You are correct that differences in heat are already accounted for in $H$ and $H_{ice}$ (e.g. through differences in albedo). $c_{g}$ accounts for differences in energy required to melt ice versus snow. (e.g. due to differences in crystal structure and thermal conductivity). The ratio is used in order to allow melt of both snow and ice in the same time step. For short time steps, this is not necessary, but it allows the model to scale better for applications at larger time steps.**

Eqs. (14-15): Please introduce $r_f$ after eq. (14) (where it first appears) instead of after eq. (15). The units of $r_f$ do not work out in eqs. (14) and (15) (in eq. (14) it would have to be in m, while according to eq. (15) it is in $m^3$). Additionally, in eq. (14) $r_{f,i}$ should likely be $r_{f,i-1}$, otherwise there would be a circular reference?

**Thank you for this correction. We have updated the relevant equations accordingly (Eqs. 21 and 22 in the revised manuscript).**

P17L3: Stating that glacier models are commonly evaluated only for a few days (!) is probably an exaggeration – I think it is well established that multi-year evaluation periods are necessary for glaciological purposes.

**We have updated the text in Section 4 (P22L1-L5). Our point is that model assessment is often inadequate, and that there are a few ways in which this is sometimes the case.**

**Technical corrections**

There is a typo in the title ("Cryrosphere")

P4L10: [$f_s$] should be [$f_m$], in the units again m instead of $m^{-1}$

P4L24: ti → to • P5L1: latent heat → latent heat of fusion

P5L3: i.e → i.e.

P5L29: (Simpson et al., 2002) → Simpson et al. (2002)

Table 1: $\alpha_0$ (from the table heading) never actually appears in the table

P9L3: $m^-2$ → $m^{-2}$

P9L24: I would move this sentence ("... where the negative bounds on CC is  zero") a few lines up (after eq. (9)).

P10L17: "the the"

P11L20: $c_t$ should probably be $t_l$?

P15L6: their → there

P18L21-22: "has the predictive skill" → "has the best predictive skill"? (2x)

Table 8 (heading): vise → vice

**Thank you, we have fixed each of the above technical corrections.**

**REFERENCES**

Beamer, J. P., Hill, D. F., Arendt, A., & Liston, G. E. (2016). High-resolution modeling of coastal freshwater discharge and glacier mass balance in the Gulf of Alaska watershed. *Water Resources Research*, *52*(5), 3888–3909. http://doi.org/10.1002/2015WR018457

Hock, R. (1999). A distributed temperature-index ice- and snowmelt model including potential direct solar radiation. Journal of Glaciology, 45(149), 101–111.

Hydrology Handbook (1996). ASCE Publications, Reston, VAAtaie-Ashitani B, Volker RE, Lockington DA (2001) Tidal effects on groundwater dynamics in unconfined aquifers. Hydrol Processes, 15, 655669.

Parajka, J., & Blöschl, G. (2008). The value of MODIS snow cover data in validating and calibrating conceptual hydrologic models. Journal of Hydrology, 358(3–4), 240–258. http://doi.org/10.1016/j.jhydrol.2008.06.006

---

## Author Response (AR2)

Response to Reviewer and Editor (Dr. Valentina Radic), TCD Manuscript

"How Much Cryosphere Model Complexity is Just Right? Exploration Using the Conceptual Cryosphere Hydrology Framework"

Mosier et al.

25 August 2016

We thank the anonymous reviewer and Dr. Valentina Radic (editor) for their reviews. The requested minor revisions were very helpful and straightforward to implement. In our previous letter responding to reviewer comments, we stated that we would make the CCHF code available on GitHub if this manuscript were accepted for publication. We have gone ahead and uploaded CCHF version 1 to GitHub under `thomasmosier/CCHF', and added two references in the manuscript text to point interested parties to this repository.

We would also like to note that in addition to addressing the reviewer's comments, we have also edited the manuscript's abstract. These changes do not impact the underlying technical content of the abstract; the purpose of the abstract changes was to improve the motivation for the work and clarity.

Below are our responses to the anonymous reviewer's comments. Following these comments is a version of our manuscript highlighting all of the changes that we made to the manuscript (produced using 'latexdiff' as recommended by the reviewer). The reviewer's original comments are in plain text and our author responses are in **bold**. Text added to the manuscript is provided in ***bold italic*** font. References to specific lines have the format PxLy, where the x refers to the page number and y to the line. Thank you for your consideration of our revised manuscript.

Kind Regards,

Thomas M. Mosier, David F. Hill, and Kendra V. Sharp

**REVIEWER:**
The authors have substantially revised their analysis and the manuscript. The MODIS evaluation procedure is significantly improved by comparing the spatial SCA patterns instead of spatial averages and by excluding glacier pixels from the analysis. Also the one-watershed validation assessment is very insightful due to the contrasting results with the proxy-basin tests. All of my initial comments have been addressed and I recommend publishing the manuscript in TC subject to a few technical corrections:

**We would like to thank this reviewer for both of their reviews and for their recognition of the improvements that we made to our manuscript in response to their previous comments.**

- Abstract, L1-2: possibly avoid the word "scenarios" (2x), as I assume you refer to the changing climate itself and not to climate (change) scenarios
**This is an important point. We have removed the word "scenario" from our abstract.**

- P12 L8: c_hoc,s and c_hoc,i do not appear in eq. (9)
**We have updated Eq. 9 to include these parameters and the text following it (P12L7-P12L8) to better explain these fitting parameters.**

- P13 L13: "where the negative bound on w_c is zero" -> possibly move this part a few lines up (after eq. (14) where w_c is introduced); replace "negative bound" with "lower bound"
**We have moved this text to P13L9-P13L10 and replaced "negative bound" with "lower bound".**

- P20 L14: missing parentheses around citations
**We have updated this reference (P20L14-P20L15).**

- P20 L29: typo in "cyrosphere"
**We have corrected this typo (P20L29).**

- Tables B3, B4, B5: "one minus the PBE or KGE score" is a bit misleading (I interpret this that one minus PBE or one minus KGE are shown, whereas in fact - if I understand correctly - one minus PBE, or KGE (without one minus) are shown?) - possibly reformulate this.
**You are correct in your interpretation of the statistics that we have provided in Tables B3, B4, and B5. We have updated the captions for these tables in the hopes of clarifying the description. The relevant revised text is "*one minus the PBE score for the 'SCA' columns and the KGE score for the 'Flow' and 'Stake' columns*".**

General remark: the parts that were changed in the manuscript are highlighted in the revised version, but to determine what exactly had been changed I had to compare it with the original manuscript (additionally, some changed parts are not highlighted, e.g. P2 L18-34, P16 L13-18, P18 L2-10). For the future I would recommend to use latexdiff or similar tools to mark the changes.
**Thank you for suggesting that we use `latexdiff' - It is a very useful tool! We used it to produce the attached highlighted version of our revised manuscript.**

[revised manuscript text omitted]